# Octopamine integrates the status of internal energy supply into the formation of food-related memories

**Michael Berger, Michèle Fraatz, Katrin Auweiler, Katharina Dorn, Tanna El Khadrawe, Henrike Scholz\***

Department of Biology, Institute for Zoology, University Köln, Köln, Germany

**Abstract** The brain regulates food intake in response to internal energy demands and food availability. However, can internal energy storage influence the type of memory that is formed? We show that the duration of starvation determines whether *Drosophila melanogaster* forms appetitive short-term or longer-lasting intermediate memories. The internal glycogen storage in the muscles and adipose tissue influences how intensely sucrose-associated information is stored. Insulin-like signaling in octopaminergic reward neurons integrates internal energy storage into memory formation. Octopamine, in turn, suppresses the formation of long-term memory. Octopamine is not required for short-term memory because octopamine-deficient mutants can form appetitive short-term memory for sucrose and to other nutrients depending on the internal energy status. The reduced positive reinforcing effect of sucrose at high internal glycogen levels, combined with the increased stability of food-related memories due to prolonged periods of starvation, could lead to increased food intake.

**\*For correspondence:**
henrike.scholz@uni-koeln.de

**Competing interest:** The authors declare that no competing interests exist.

## eLife assessment

This **important** study dissects the role of octopamine in the interplay between internal energy homeostasis, food intake, and food-related memories. The **solid** experimental evidence will shed additional light on previously published work and should be of interest to the growing community of biologists interested in how internal state shapes behavior, including decision-making processes, learning and memory.

## Introduction

To ensure survival, the internal energy status of an animal needs to be adjusted to the energy expenditure of the organism and the availability of external food. Increased storage of energy correlates with increased food intake in the past. Dysregulation of food intake can lead to diseases such as obesity and diabetes. Sucrose is a carbohydrate enriched in Western diets, and the breakdown product of sucrose – glucose – can be stored in the organism as glycogen, primarily in the liver and muscles. Elevated glycogen levels are a hallmark of glycogen storage diseases that are accompanied by defects in the liver, muscles, and brain (*Ellingwood and Cheng, 2018*). Similar to vertebrates, the fruit fly *Drosophila melanogaster* uses glucose as its primary energy source and stores glycogen primarily in the muscles – the main site of energy expenditure – and the fat body – the equivalent of the vertebrate liver (*Gáliková and Klepsatel, 2023*; *Wigglesworth, 1949*). As in vertebrates, glycogen levels are also found in the brain (*Yamada et al., 2018*).

In *Drosophila*, the monoamine octopamine – functionally related to noradrenalin in vertebrates – is involved in the regulation of energy homeostasis. In tyramine-β-hydroxylase (*Tβh*) mutants lacking the

**eLife digest** Deciding what and how much to eat is a complex biological process which involves balancing many types of information such as the levels of internal energy storage, the amount of food previously available in the environment, the perceived value of certain food items, and how these are remembered.

At the molecular level, food contains carbohydrates that are broken down to produce glucose, which is then delivered to cells under the control of a hormone called insulin. There, glucose molecules are either immediately used or stored as glycogen until needed. Insulin signalling is also known to interact with the brain's decision-making systems that control eating behaviors; however, how our brains balance food intake with energy storage is poorly understood. Berger et al. set out to investigate this question using fruit flies as an experimental model.

These insects also produce insulin-like molecules which help to relay information about glycogen levels to the brain's decision-making system. In particular, these signals reach a population of neurons that produce a messenger known as octopamine similar to the human noradrenaline, which helps regulate how much the flies find consuming certain types of foods rewarding. Berger et al. were able to investigate the role of octopamine in helping to integrate information about internal and external resource levels, memory formation and the evaluation of different food types.

When the insects were fed normally, increased glycogen levels led to foods rich in carbohydrates being rated as less rewarding by the decision-making cells, and therefore being consumed less. Memories related to food intake were also short-lived – in other words, long-term 'food memory' was suppressed, re-setting the whole system after every meal.

In contrast, long periods of starvation in insects with high carbohydrates resources produced a stable, long-term memory of food and hunger which persisted even after the flies had fed again. This experience also changed their food rating system, with highly nutritious foods no longer being perceived as sufficiently rewarding. As a result, the flies overate.

This study sheds new light on the mechanisms our bodies may use to maintain energy reserves when food is limited. The persistence of 'food memory' after long periods of starvation may also explain why losing weight is difficult, especially during restrictive diets. In the future, Berger et al. hope that this knowledge will contribute to better strategies for weight management.

neurotransmitter octopamine, the glucose and trehalose concentrations in the hemolymph change less upon starvation, and the life span is extended (*Damrau et al., 2017*; *Li et al., 2016*). *Tβh* mutants show an increased threshold to respond to sucrose with the extension of their proboscis when sucrose is offered to their tarsi – a structure that contains gustatory receptor neurons. The reduced responsiveness to sucrose correlates with a reduced sucrose intake in *Tβh* mutants (*Li et al., 2016*; *Scheiner et al., 2014*). Octopaminergic neurons potentiate the response of sugar-sensing gustatory receptor neurons in satiated flies, suggesting that octopaminergic neurons regulate feeding behavior by changing the sensitivity of taste receptor neurons (*Youn et al., 2018*). The reduced responsiveness to sucrose is also thought to be responsible for the defects observed in habituation in *Tβh* mutants (*Scheiner et al., 2014*). In addition to their changes in simple forms of neuronal plasticity, *Tβh* mutants fail to form a positive association with a food reward in a classical olfactory conditioning paradigm. Flies quickly learn to associate an odorant with sucrose, and Tβh mutants do not show this positive association directly after learning (*Schwaerzel et al., 2003*). However, when a longer time after training has elapsed, memory appears (*Das et al., 2014*). The release of octopamine mediates the reinforcing effect of sweet taste in short-term memory (STM) and the anesthesia-resistant form of long-term memory (LTM) (*Burke et al., 2012*; *Huetteroth et al., 2015*; *Wu et al., 2013*). The question arises how the internal energy supply is integrated into the formation of food-related memories and feeding behavior in *Tβh* mutants.

The peptide hormone insulin regulates glycose homeostasis at the cellular level (*Saltiel and Kahn, 2001*). Insulin and its *Drosophila* counterpart, insulin-like peptides, perform their functions through well-conserved signal transduction cascades (*Chatterjee and Perrimon, 2021*; *Inoue et al., 2018*). In vertebrates, in addition to its function in fat tissue and muscles, the insulin receptor is broadly expressed in the brain and regulates neuronal plasticity (*Nakai et al., 2022*). For example, in rats, reduced insulin

receptor function in the hypothalamus results in loss of long-term potentiation and impaired spatial memory (*Grillo et al., 2015*). In *Drosophila*, the insulin receptor substrate Chico is required for the development of mushroom bodies, a brain region required for learning and memory, and loss of Chico results in learning defects (*Naganos et al., 2012*). Disrupting the function of the insulin receptor in the mushroom bodies also results in learning defects, whereas the insulin receptor and Chico are required in the ellipsoid body for protein biosynthesis-dependent LTM (*Chambers et al., 2015*). While there is evidence that insulin receptor signaling regulates cellular physiology, it is not clear how the internal energy supply of the animal influences the formation of food-related memories.

To analyze the relationship between energy status, the evaluation of a food reward, and the formation of memory and food intake, we used the olfactory associative learning and memory paradigm. Using nutrients as a positive reward allows for investigating how the internal energy status influences the evaluation of the reinforcer and in turn learning and memory. As a step to understand the interconnection between reward evaluation, nutrient intake, and the formation of food-related memories, we analyzed the function of octopamine in the regulation of internal energy homeostasis, learning and memory, and food intake.

## Results

Starvation increases learning and memory performance when using a food-related reward (*Colomb et al., 2009*). However, how the internal energy status is integrated into the evaluation of the reward and the stability of food-related memories and whether there is a correlation between the perceptions of the food reward, food-related memories, and food intake are not clear.

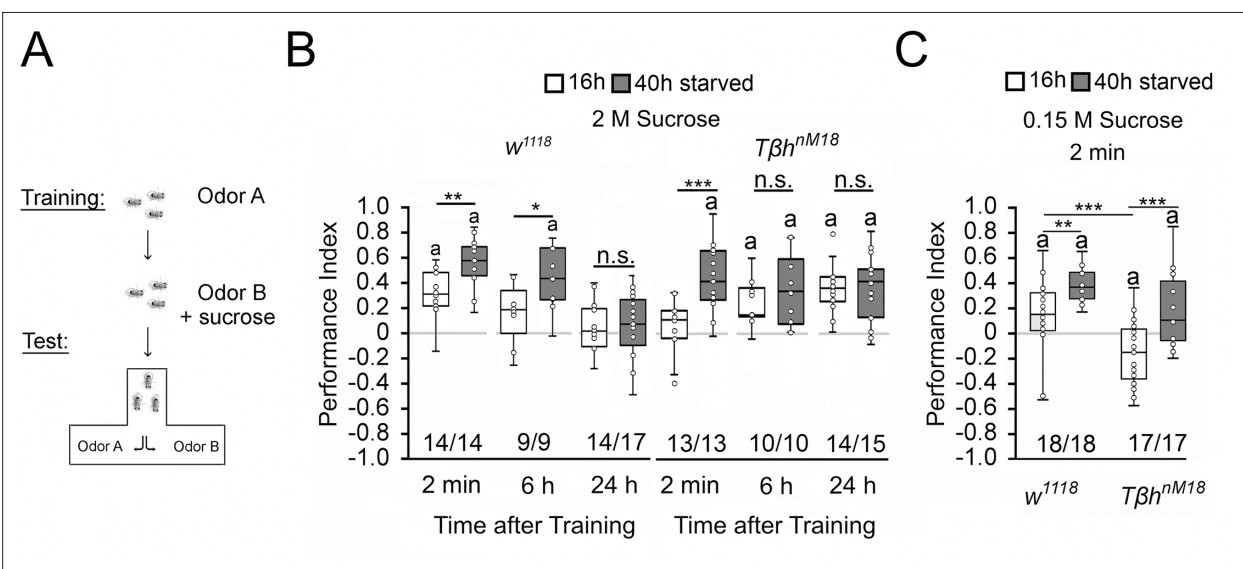

**Figure 1.** Starvation influences the strength of memory. (**A**) Appetitive olfactory learning and memory paradigm. (**B**) Appetitive 2 min short-term memory (STM), 6 hr and 24 hr memory of $w^{1118}$ or $T\beta h^{nM18}$ that were starved either for 16 hr (white bars) or 40 hr (dark gray bars) before the training. 2 M sucrose was used as reward. (**C**) 0.15 M sucrose was used as reinforcer. Prolonged starvation increases memory performance. Prolonged starvation from 16 hr to 40 hr leads to the formation of STM in $T\beta h^{nM18}$. Independent of the duration of starvation, 6 hr after training, memory appeared in the mutants. (**C**) $T\beta h^{nM18}$ mutants starved for 16 hr developed an aversive STM to 0.15 M sucrose. After 40 hr of starvation, the mutants developed an appetitive STM. Numbers below box plots indicate pairs of reciprocally trained independent groups of male flies. The letter 'a' marks a significant difference from random choice as determined by a one-sample sign test (p<0.05). Student's *t*-tests were used to determine differences between two groups. For differences between more than two groups, one-way ANOVA with Tukey's post hoc Honest Significant Difference (HSD) test was used. (*p<0.05; **p<0.01; ***p<0.001).

The online version of this article includes the following source data for figure 1:

**Source data 1.** The raw data and sensory acuity data of *Figure 1* and related supplement.

# The duration of hunger influences the strength and stability of memories

To investigate whether changes in internal energy storage influence food-related memories, we used the olfactory associative learning and memory paradigm using sucrose as a food-related reinforcer (*Schwaerzel et al., 2003*). In the paradigm, flies learn to associate an odorant with sucrose and are later tested on whether they remember the association. A prerequisite for this association to be made is that the animals are hungry. During the training, starved flies were exposed for 2 min to the unconditioned odorant, 4-methylcyclohexanol (MCH), followed by a 2 min exposure to a second odorant, 3-octanol (3-OCT), that was paired with the reward of 2 M sucrose. After the training, flies were given a choice between the two odorants (*Figure 1A*). Normally, flies prefer the rewarded odorant already 2 min after the training. To avoid a bias towards one odorant, in a reciprocal experiment, the first odorant was rewarded. The performance index presents the average positive association between the reward and both odorants and describes whether information is learnt and remembered. To ensure that observed differences in learning and memory were not due to changes in odorant perception, odorant evaluation, or sucrose sensitivity, different fly populations of the same genotypes were tested for their odorant acuity, odorant, and sucrose preference (*Figure 1—source data 1*). The flies of the different genotypes sensed the odorants and evaluated them as similar salient in comparison. This is important to avoid bias where flies have to choose between the two odorants after training. They also sensed sucrose. We next determined whether the differences in sucrose preference influence sucrose intake during training. Therefore, we measured the intake of food-colored sucrose of starved flies in the behavioral set up. After 2 min, we evaluated whether there was dyed food in the abdomen of the fly (*Figure 1—source data 1*). Flies of both genotypes fed sucrose within 2 min and there was no difference between $w^{1118}$ and $T\beta h^{nM18}$ flies.

To alter the internal energy status, flies were starved either for 16 hr or for 40 hr before the training (*Figure 1*). We used 3- to 5-day-old male flies to minimize differences in body weight and control for differences in food preferences. We included $T\beta h^{nM18}$ mutants lacking octopamine in the experiments since they showed defects in energy metabolism and sucrose reward learning (*Li et al., 2016*; *Schwaerzel et al., 2003*). Flies that were starved for 16 hr established a positive association between the odorant and the reward. They remembered this association for 2 min, but not for 6 hr or 24 hr. Consistent with previous results, learning and memory performance was significantly improved by prolonged starvation (*Colomb et al., 2009*). The memory was also remembered for longer since it was still detectable 6 hr after training. The $T\beta h^{nM18}$ mutants starved for 16 hr did not show 2 min memory but emerged memory 6 hr later that lasted up to 24 hr (*Das et al., 2014*; *Schwaerzel et al., 2003*). In contrast, $T\beta h^{nM18}$ mutants remembered the rewarded odorant after only 2 min following starvation extended to 40 hr. The memory was still detectable after 6 hr and 24 hr (*Figure 1B*). We repeated similar experiments using a lower concentration of 0.15 M sucrose as reinforcer (*Figure 1C*). In contrast to the experiments using 2 M sucrose as a reward, 16 hr-starved $T\beta h^{nM18}$ mutants form a negative association with 0.15 M sucrose directly after the training, which turns into a positive association when the flies were starved longer 40 hr. Again, prolonged starvation significantly increased learning and memory performance in controls and $T\beta h^{nM18}$ mutants. Thus, the length of starvation and the reinforcer strength influence appetitive memory strength.

Next, we investigated how starvation affects the stability of memory. Protein synthesis-dependent LTM is still labile immediately after training and can be blocked by 4°C cold-shock anesthesia immediately after training. After consolidation, LTM is cold-shock-resistant and insensitive to a cold shock 1 hr before the test (*Krashes and Waddell, 2008*). To investigate how hunger influences the stability of food-related memories, we administered cold-shock anesthesia immediately after the training and 1 hr before the test and analyzed the effect on memory formation in differently starved control and $T\beta h^{nM18}$ flies (*Figure 2A*). We reduced the time for the memory test after training from 6 hr to 3 hr to be able to detect memory in 16 hr-starved flies. The memory was completely abolished by a cold shock directly after training or shortly before the test. Prolonging the starvation period resulted in memory that was sensitive to cold shock directly after training but not shortly before the test. In contrast, 16 hr-starved $T\beta h^{nM18}$ mutants developed memory that was sensitive to cold shock directly after training and cold-shock-insensitive shortly before the test. The results, together with the observed memory 24 hr later of $T\beta h^{nM18}$ after the same training (*Figure 1B*), support that $T\beta h^{nM18}$ mutant develop LTM. Longer periods of starvation in $T\beta h^{nM18}$ mutants resulted in anesthesia-resistant LTM.

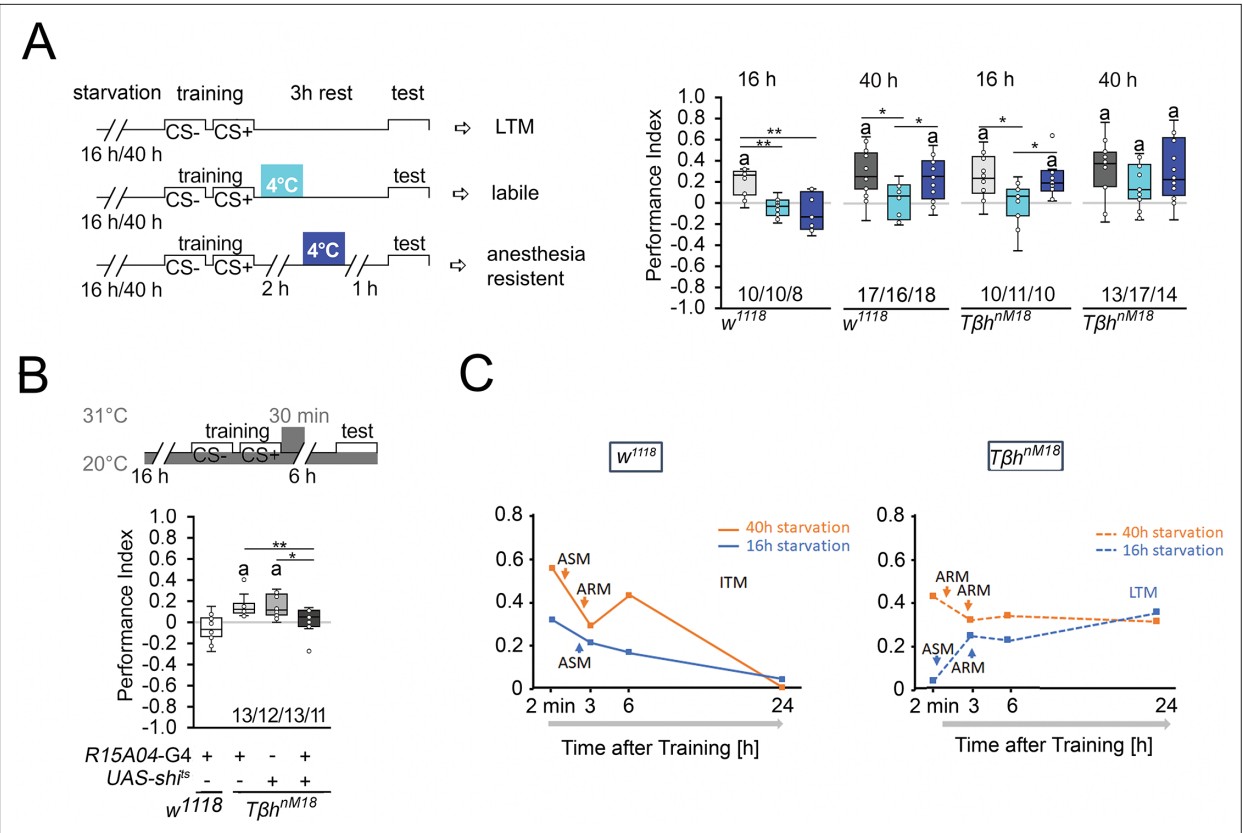

**Figure 2.** Starvation influences the type of memory. (**A**) Appetitive short-term memory (STM) training with 2 M sucrose and cold shock directly or 2 hr after training. Mildly starved control flies exhibit an appetitive memory sensitive to cold shock 3 hr after training. Severely starved control flies develop a memory that is initially sensitive to cold shock, but becomes insensitive after 2 hr. This phenotype is shared with mildly starved $T\beta h^{nM18}$ mutants. Prolonged starvation in the mutants shifts memory to cold-shock-insensitive memory. (**B**) To block neuronal activity and the formation of LTM, a 30 min heat shift was applied immediately after training to flies expressing a temperature-sensitive *shibire* transgene under the control of the R1504-Gal4 driver. The block results in $T\beta h^{nM18}$ mutants losing LTM. Numbers below box plots indicate pairs of reciprocally trained independent groups of male flies. The letter 'a' marks a significant difference from random choice as determined by a one-sample sign test (p<0.05). One-way ANOVA with Tukey's post hoc Honest Significant Difference (HSD) test or for data in (**A**) and Kruskal–Wallis followed by post hoc Dunn's test and Bonferroni correction was used to determine differences in (**B**). *p<0.05; **p<0.01. (**C**) Model summarizing memory performance of control and mutant flies that were either starved for 16 hr (blue line) or 40 hr (orange line). The dots present the average of the data presented in *Figures 1* and Figure 2. ASM: anesthesia-sensitive memory; ARM: anesthesia-resistant memory; ITM: intermediate memory; LTM: long-term memory. Source data see *Figure 2—source data 1*.

The online version of this article includes the following source data and figure supplement(s) for figure 2:

**Source data 1.** The raw data and sensory acuity data of *Figure 2* and related supplement.

**Figure supplement 1.** Controls for heat shift experiment in *Figure 2B*.

To further confirm that the observed memory in 16 hr-starved $T\beta h^{nM18}$ is indeed LTM, we wanted selectively inhibit LTM in $T\beta h^{nM18}$ mutants and analyze their food-related memory (*Figure 2B*). We analyzed memory performance 6 hr after training as control flies show no memory at this time point (*Figures 1B and 2B*). The R15A04-Gal4 driver targets dopaminergic neurons of the PAM cluster, which are specifically required for appetitive LTM but not STM (*Yamagata et al., 2015*). Blocking the function of these dopaminergic neurons immediately after training using a temperature-sensitive *shibire* transgene (UAS-shi<sup>ts</sup>) and a 30 min heat pulse of 31°C resulted in loss of memory in $T\beta h^{nM18}$ mutants (*Figure 2B*). Control flies carrying one copy of the Gal4 transgene showed no memory. Without heat shock, the $T\beta h^{nM18}$ mutants carrying the UAS-shi<sup>ts</sup>, the R15A04-Gal4, or both transgenes developed memory (*Figure 2—figure supplement 1*). Thus, although they do not show memory immediately after training, $T\beta h^{nM18}$ mutants that were briefly starved develop LTM.

These results, together with the course of memory loss, show that lightly hungry flies form a memory that lasts up to 3 hr and is sensitive to anesthesia (*Figure 2D*). Since no memory can be observed 24 hr later, this indicates that it is an anesthesia-sensitive intermediate memory (ITM). Prolonged starvation

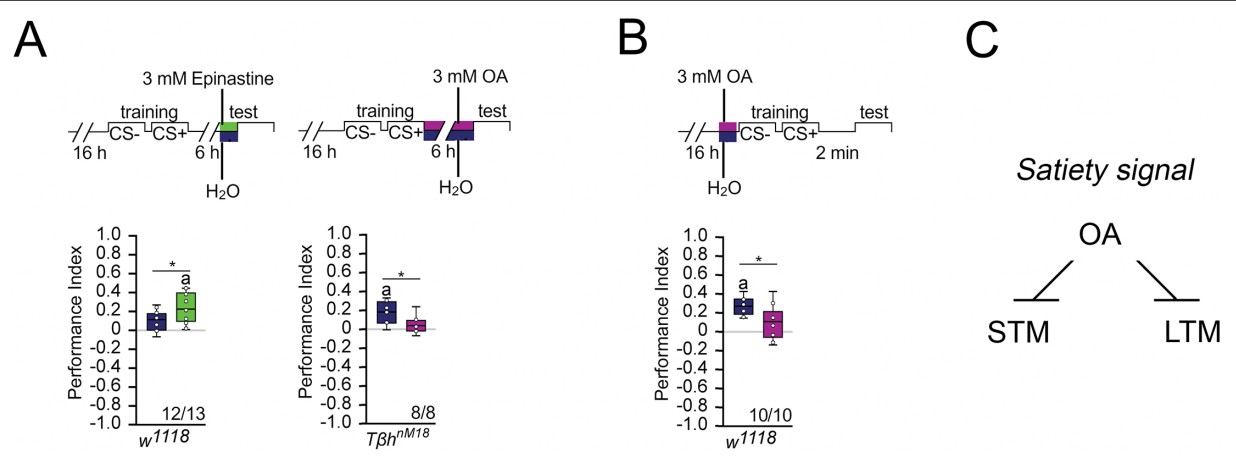

**Figure 3.** Octopamine suppresses memory. (**A**) Feeding 3 mM of the octopamine receptor antagonist epinastine for 1 hr after training resulted in memory 6 hr later in $w^{1118}$ flies. Feeding 3 mM octopamine for 6 hr after training suppresses long-term memory (LTM) in $T\beta h^{nM18}$. (**B**) A 3 mM octopamine feeding pulse 30 min before training inhibits short-term memory (STM) in $T\beta h^{nM18}$ mutants. Controls were water-fed. (**A, B**) Flies were starved for 16 hr and 2 M sucrose was used as reinforcer. The letter 'a' marks a significant difference from random choice as determined by a one-sample sign test (p<0.05). Student's $t$-tests were used to determine differences between two groups (*p<0.05). Numbers below box plots indicate pairs of reciprocally trained independent groups of male flies. (**C**) Model for memory suppression.

The online version of this article includes the following source data and figure supplement(s) for figure 3:

**Source data 1.** The raw data of *Figure 3* and related supplement.

**Figure supplement 1.** Octopamine but not tyramine suppresses food intake.

results in anesthesia-resistant ITM (*Figure 2D*). In contrast, mildly starved $T\beta h^{nM18}$ mutants form a LTM that is sensitive to anesthesia-sensitive and detectable 24 hr after the training. Prolonged starvation in $T\beta h^{nM18}$ mutants leads to anesthesia-resistant LTM (*Figure 2D*).

Thus, the longer the starvation lasts, the more stable the memory becomes. Depending on the duration of starvation after the same training phase, the animals first form an STM, then anesthesia-sensitive or insensitive ITM. Mildly starved $T\beta h^{nM18}$ mutants do not form an STM, but a LTM and after prolonged starvation, short-term and anesthesia-resistant memory.

## Octopamine is a negative regulator of memory

Since the $T\beta h^{nM18}$ mutants lack the neurotransmitter octopamine and form LTM, we next investigated whether octopamine normally suppresses LTM. To do so, we first blocked the function of octopamine receptors in controls immediately after the training by feeding the octopamine antagonist epinastine for 1 hr and analyzed memory 5 hr later (*Figure 3A*). If the octopamine receptor function is required for LTM, prolonged memory should also occur in control flies that normally show no memory after 16 hr of starvation. This was the case. Consistent with the idea that octopamine is a negative regulator of LTM, feeding octopamine to $T\beta h^{nM18}$ mutants immediately after training blocked LTM (*Figure 3A*). To analyze whether octopamine is also able to block STM, we fed octopamine prior to training to control flies (*Figure 3B*). A short pulse of octopamine before the training inhibits STM.

We were wondering whether octopamine could reduce the appetite in starved animals and thereby reducing the value of the food reward. To investigate this, we fed starved flies with a short octopamine pulse and assayed their sucrose intake after 3 hr (*Figure 3—figure supplement 1*). Flies fed with octopamine showed a significant reduction of sucrose intake, but not flies fed with a similar amount of tyramine or a tenfold increase in tyramine concentration. Therefore, it is possible that increased octopamine content influences how food is evaluated. This is consistent with a function of octopamine as a signal for food or high internal energy levels. Taken together, octopamine is a negative regulator of appetitive LTM and STM (*Figure 3C*).

## Starvation influences sucrose consumption preference

Starvation reduces the internal energy storage. The reduction could lead to a reassessment of the external food cues and increased food consumption to restore the energy supplies. First, we

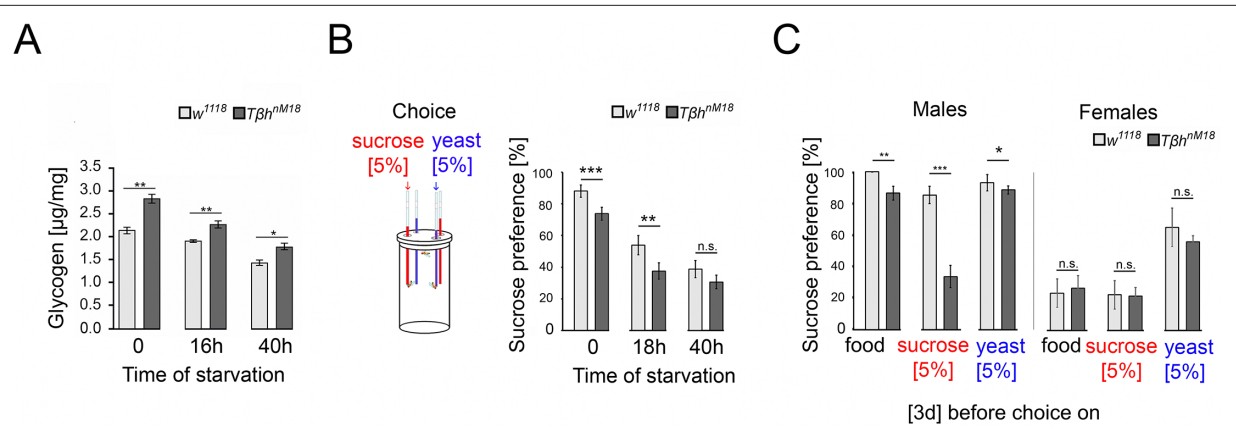

**Figure 4.** Elevated glycogen levels correlate with reduced sucrose preference. (**A**) Analysis of whole-body glycogen levels in $w^{1118}$ and $T\beta h^{nm18}$ flies. In $T\beta h^{nM18}$ flies, glycogen content is significantly higher than those in $w^{1118}$ flies under similar starvation conditions. N = 3 groups of five male flies. (**B**) Flies were starved 18 hr or 40 hr before food intake was measured for 24 hr. Flies chose between 5% sucrose and 5% yeast. The preference was determined. All flies showed a significant preference for sucrose consumption. Starvation reduced the preference. $T\beta h^{nM18}$ showed a significantly reduced preference for sucrose in comparison to control flies, but not after 40 hr starvation. N = 20–26 groups of eight flies. (**C**) Feeding male flies for 3 d on standard fly food (food), 5% sucrose, or 5% yeast resulted in control flies preferring to consume sucrose. Male $T\beta h^{nm18}$ mutants fed normal food, sucrose, and yeast showed a significant reduction in sucrose preference. Female flies of controls and mutants did not differ in their preferences. N = 14–28 groups of eight flies. To determine differences between two groups, the Mann–Whitney $U$ test was used. *$p<0.05$; **$p<0.01$; ***$p<0.001$.

The online version of this article includes the following source data for figure 4:

**Source data 1.** Raw data related to *Figure 4*.

investigated how starvation reduces the glycogen storage in whole animals (*Figure 4A*). In controls and $T\beta h^{nM18}$ mutants, starvation reduces the glycogen level. However, non-starved $T\beta h^{nM18}$ mutant males had significantly higher glycogen levels at baseline. After 40 hr of starvation, the glycogen levels of $T\beta h^{nM18}$ mutants were still higher than those of controls that were starved for the amount of time.

Adult $T\beta h^{nM18}$ mutants have a reduced sucrose intake (*Li et al., 2016*; *Scheiner et al., 2014*). The assessment of external food sources might be reflected in the choice of food. To examine whether hunger influences the evaluation of an external food stimulus, we starved flies and determined the preference for consuming sucrose over protein-enriched food using the capillary feeder assay (*Figure 4B*). Non-starved flies reared on standard fly food preferred sucrose over protein-enriched food, but $T\beta h^{nM18}$ mutants to a significantly lesser extent than controls. Also, 18 hr of starvation resulted in a decreased sucrose preference in controls and $T\beta h^{nM18}$ mutants, but to a significantly lower level in $T\beta h^{nM18}$ mutants. Extending the starvation period to 40 hr resulted in a similar sucrose consumption in control and mutant flies.

To further investigate whether $T\beta h^{nM18}$ mutants integrate their internal glycogen storage into their feeding preference, we deprived male flies of sucrose- or protein-enriched food by feeding them 5% sucrose, 5% yeast, or standard fly food for 3 d. The food preference was analyzed after deprivation (*Figure 4C*). As controls, we included mated females in our analysis as they have different nutritional requirements due to their mating status (*Ribeiro and Dickson, 2010*; *Vargas et al., 2010*). Male flies fed standard food for 3 d strongly preferred sucrose over protein-enriched food, with $T\beta h^{nM18}$ mutants showing a significant reduction. Yeast-deprived and sucrose-deprived $T\beta h^{nM18}$ mutant showed a significantly reduced preference for sucrose. In mated females, no differences were observed in food preference between controls and the $T\beta h^{nM18}$ mutant. In conclusion, the reduced preference for sucrose consumption correlates in $T\beta h^{nM18}$ with increased glycogen levels. Furthermore, $T\beta h^{nM18}$ mutants can sense the reduction of specific internal energy stores and change their food preferences accordingly.

## Internal glycogen storage influences sucrose-related memories

Since memory performance increases when the internal energy supply is reduced by starvation, we wanted to investigate whether the internal energy supply influences memory performance. In *Drosophila*, glycogen is mainly found in the fat bodies – the major energy storage organ – and the

muscles, a major site of energy expenditure (*Wigglesworth, 1949*). Glycogen synthase and glycogen phosphorylase control glycogen content in the body (*Figure 5A*). Knockdown of glycogen synthase using $GlyS^{HMS01279}$-RNAi efficiently reduced glycogen levels in larvae, while knockdown of glycogen phosphorylase using $GlyP^{HMS00032}$-RNAi efficiently increased glycogen levels (*Yamada et al., 2018*). We altered glycogen levels in the muscles using the *mef2*-Gal4 driver (*Ranganayakulu et al., 1998*) and/ or the fat bodies using the FB-Gal4 driver (*Grönke et al., 2003*) and analyzed the effect of altered glycogen levels on STM (*Figure 5*). We used PAS staining to confirm the down- or upregulation of glycogen in larval muscles or fat bodies, respectively (*Figure 5*; *Yamada et al., 2018*). In addition, we quantified the glycogen levels in the bodies of adult flies (*Figure 5—figure supplement 1*). As we could not always detect differences in glycogen content in whole flies, we also analyzed the glycogen level in the muscle-rich thorax or fat body-rich abdomen (*Figure 5—figure supplement 1*). The knockdown of glycogen phosphorylase increased glycogen levels, and the knockdown of glycogen synthase reduced glycogen levels.

Increasing glycogen levels in the muscles had no effect on the STM of the 16 hr-starved flies, but decreasing glycogen levels significantly improved memory performance (*Figure 5B*). Increasing or decreasing glycogen levels in the fat bodies had no effect on memory performance (*Figure 5C*). When the muscles and fat bodies glycogen levels were significantly increased, flies showed a reduced memory to odorants paired with sucrose. An increase in memory performance was observed when glycogen levels were significantly reduced in both tissues (*Figure 5D*). Recently, it has been shown that energy metabolism in mushroom bodies is important for the formation of LTM (*Plaçais et al., 2017*). To analyze the function of the mushroom bodies, the expression of $GlyS^{HMS01279}$-RNAi under the control of the Mef2-Gal4 driver was repressed in the mushroom bodies using the mb247-Gal80 driver (*Krashes et al., 2007*). The memory was still increased (*Figure 5—figure supplement 2*). Conversely, the reduction of $GlyS^{HMS01279}$-RNAi using the mb247-Gal4 driver targeting the mushroom bodies (*Zars et al., 2000*) did not alter STM (*Figure 5—figure supplement 2*). Thus, low levels of glycogen in the muscles after starvation positively influence appetitive STM, whereas high levels of glycogen in the muscles and fat body decrease reduce STM.

## Internal glycogen levels reduce sucrose-related memories in *Tβh* mutants

The increased glycogen levels in $Tβh^{nM18}$ mutants could be responsible for the reduced STM. To determine whether the reduction in glycogen levels in the muscles or fat bodies restores STM, we expressed $GlyS^{HMS01279}$-RNAi under the control of the *mef2*-Gal4 or FB-Gal4 driver in $Tβh^{nM18}$ mutants and analyzed STM (*Figure 6A*). Neither the reduction in the muscles nor the reduction in fat bodies of $Tβh^{nM18}$ mutants improved STM. Only when glycogen was reduced in both tissues did the $Tβh^{nM18}$ mutants show improved STM compared to controls. Thus, $Tβh^{nM18}$ mutant flies can form appetitive STM similar to controls when the energy storage is sufficiently reduced. Next, we analyzed whether male $Tβh^{nM18}$ mutants can form STM to other nutrients than carbohydrates by using a protein-enriched diet in the form of 5% yeast as a positive reinforcer (*Figure 6B*). To determine whether there is a difference in the evaluation of protein as a food source between male flies and $Tβh^{nM18}$ mutants, we measured yeast intake in non-starved and starved flies (*Figure 6—figure supplement 1*). Non-starved $Tβh^{nM18}$ mutant males have a significantly higher protein intake than controls. However, after 16 hr of starvation, the protein intake was comparable to that of control flies. When 5% yeast was used as a food reward, male $Tβh^{nM18}$ mutants showed comparable STM levels controls (*Figure 6B*).

To further analyze the influence of the internal energy status on memory performance, we took advantage of the observation that female flies have different nutritional requirements depending on their mating status (*Ribeiro and Dickson, 2010*; *Vargas et al., 2010*). Virgin females showed a higher consumption preference for sucrose than mated females when given the choice between yeast and sucrose (*Figure 4C*), demonstrating a difference in internal energy demands and suggesting a difference in valence for different diets. We tested virgin and mated female flies of controls and $Tβh^{nM18}$ mutants for STM with 2 M sucrose as reinforcer (*Figure 6C*). Virgin females remembered the odorant paired with sucrose significantly better than mated females. This was also true for $Tβh^{nM18}$ mutant females. Thus, the internal energy status influences how the reward is evaluated in learning and memory.

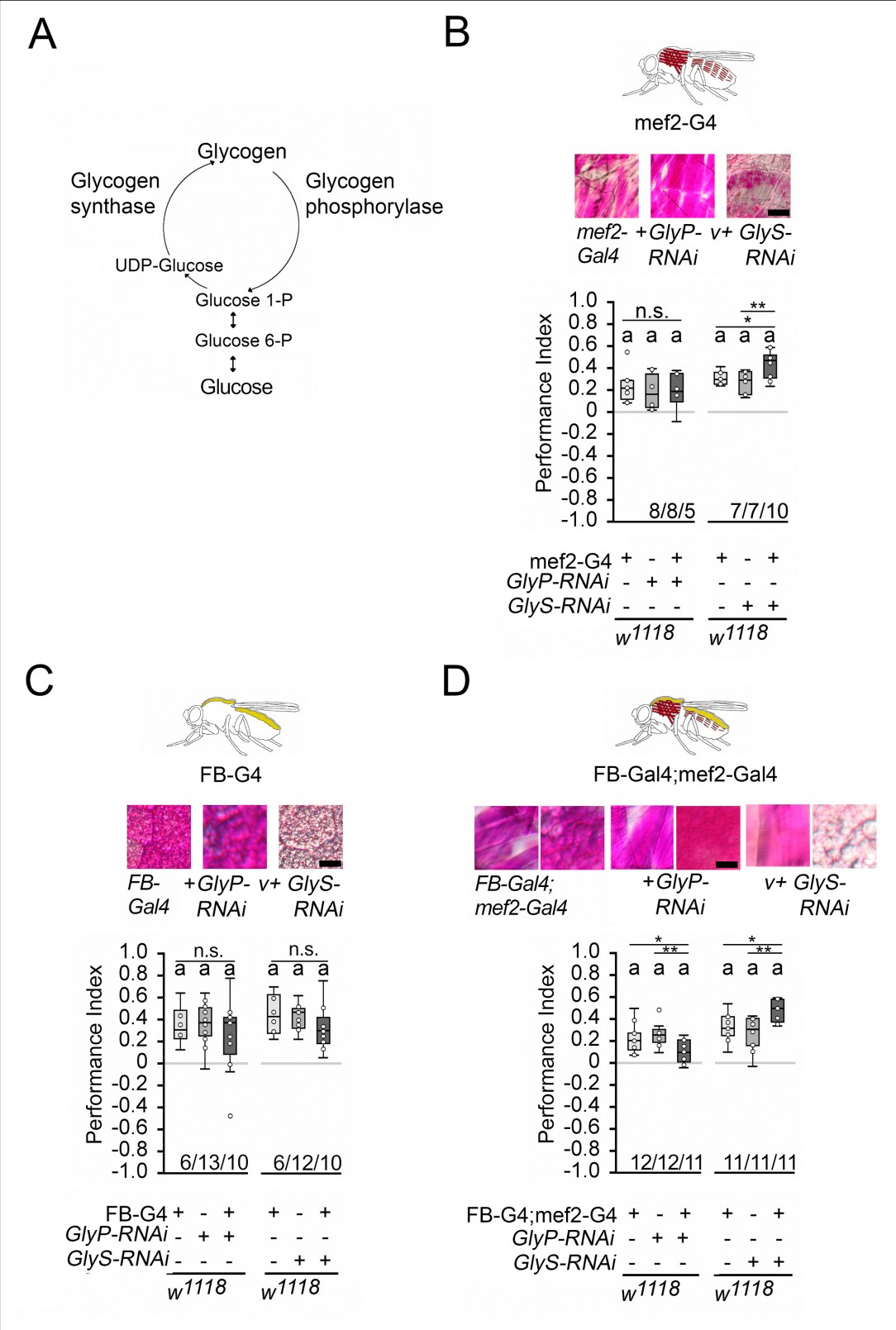

**Figure 5.** Carbohydrate storage influences appetitive short-term memory (STM). (**A**) Schemata of glycogen synthesis. The expression of *GlyP-RNAi* reduced glycogen phosphorylase and increased glycogen levels, whereas *GlyS-RNAi* reduced glycogen synthase and decreased glycogen levels in target tissues (*Figure 3*). (**B–D**) PAS was used to visualize glycogen levels in larval muscle or fat bodies. (**B**) Increases in glycogen in the muscles have no effect on STM, whereas reduced muscle glycogen increases appetitive STM. (**C**) Increased or decreased glycogen levels in the fat bodies did not

*Figure 5 continued on next page*

*Figure 5 continued*

interfere with STM. (**D**) A combined increase in glycogen in muscles and fat bodies reduced STM, and a decrease in glycogen increased STM. Flies were starved for 16 hr and 2 M sucrose was used as reinforcer. Numbers below box plots indicate one pair of reciprocally trained independent fly groups. The letter 'a' marks a significant difference from random choice as determined by a one-sample sign test (p<0.05). The one-way ANOVA with post hoc Tukey's Honest Significant Difference (HSD) was used to determine differences between three groups (*p<0.05; **p<0.01).

The online version of this article includes the following source data and figure supplement(s) for figure 5:

**Source data 1.** The raw data and sensory acuity data of *Figure 5* and related supplement.

**Figure supplement 1.** Glycogen level in adult flies with reduced GlyP and GlyS.

**Figure supplement 2.** The glycogen level in the mushroom bodies does not influence appetitive short-term memory (STM).

## Insulin-like signaling in octopaminergic neurons regulates STM

How is the internal energy status integrated into the reward system? Insulin-like signaling regulates glycogen levels in invertebrates and vertebrates (*Semaniuk et al., 2021*). Loss of the insulin receptor results in more circulating sugar but not increased glycogen levels (*Shingleton et al., 2005*). Thus, the insulin receptor might be a good candidate that links the internal energy level to reinforcing neurons. First, we analyzed whether the insulin receptor is expressed in octopaminergic reward neurons in the brain (*Figure 7A*). To detect the expression of an activated insulin receptor, we used an insulin

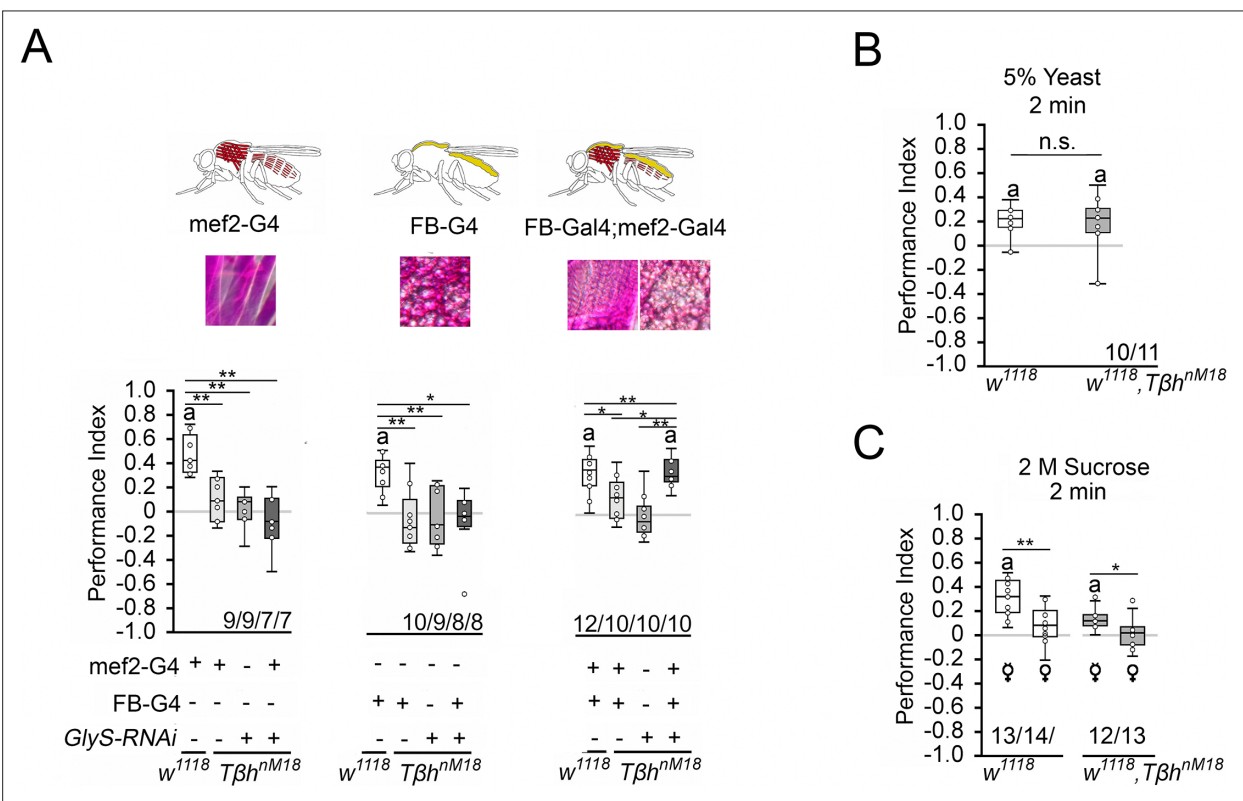

**Figure 6.** Reducing glycogen in *TβhnM18* improves appetitive short-term memory (STM). (**A**) Decreasing glycogen concentration using UAS-*GlySRNAi* in the muscles or fat bodies in *TβhnM18* mutants did not improve STM, but decreasing glycogen in both tissues improved STM to control levels. Flies were starved for 16 hr and 2 M sucrose was used as reinforcer. (**B**) 16 hr-starved *w1118* and *Tβhnm18* flies formed similar levels of appetitive STM when 5% yeast was used as a reinforcer. (**C**) 16 hr-starved virgin females of w1118 and *Tβhnm18* displayed STM, whereas mated females of both genotypes did not. Differences from random choice were determined using a one-sample sign test and marked with the letter 'a' (p<0.05). Differences between two groups were determined using Student's *t*-tests, and differences among four groups were determined with one-way ANOVA with Tukey's Honest Significant Difference (HSD) post hoc test. *p<0.05; **p<0.01. Numbers below box plots indicate one pair of reciprocally trained independent fly groups.

The online version of this article includes the following source data and figure supplement(s) for figure 6:

**Source data 1.** The raw data and sensory acuity data of *Figure 6* and related supplement.

**Figure supplement 1.** Starvation influences yeast consumption of *Tβhnm18* flies.

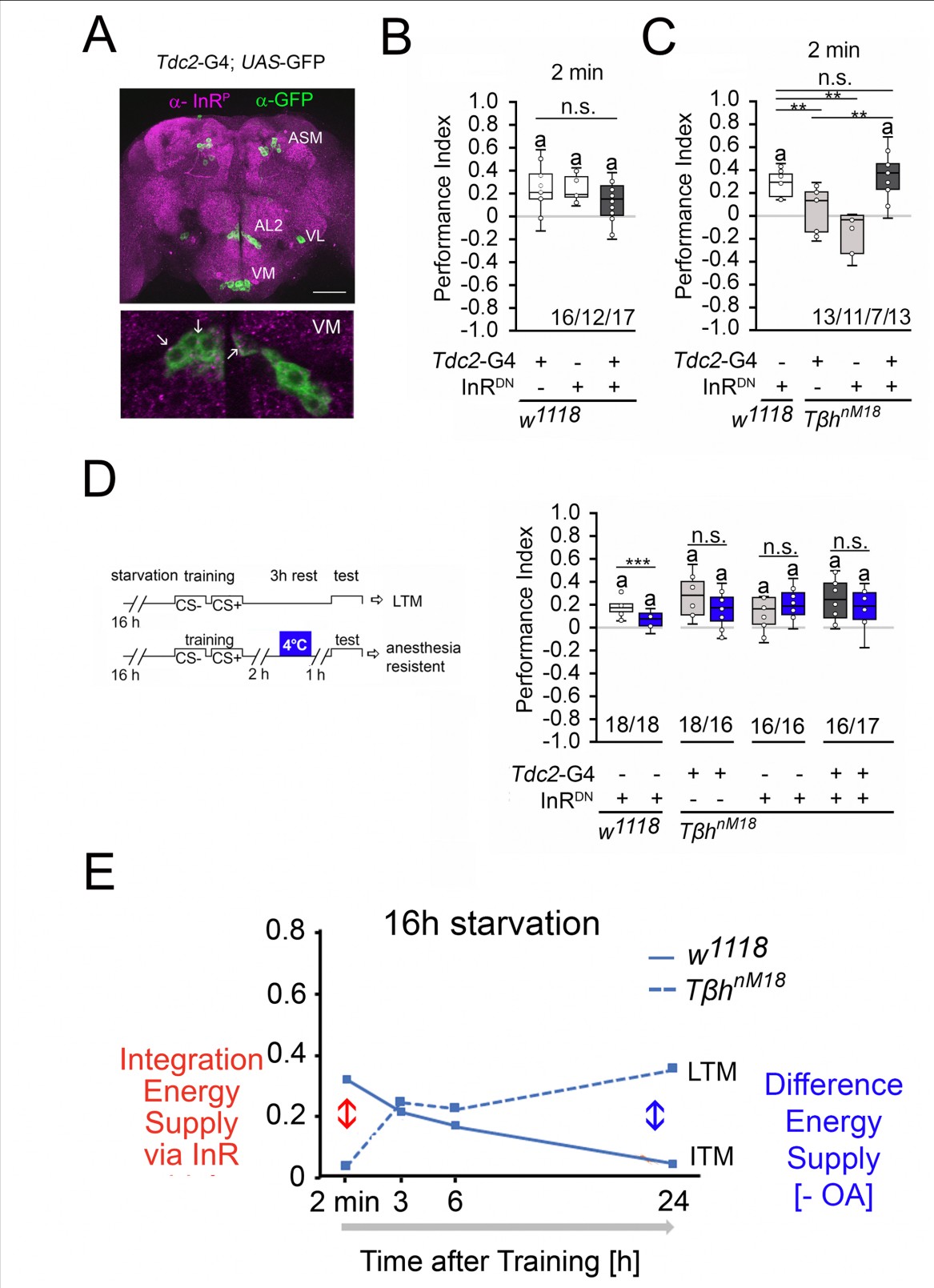

**Figure 7.** Insulin signaling in reward neurons regulates short-term memory (STM). (**A**) The activated form of the InR is expressed in punctuate manner throughout the brain (in magenta) and is also detected in octopaminergic reward neurons visualized by using the UAS-mCD8::GFP transgene under the control of the *Tdc2*-Gal4 driver (in green). (**B**) Blocking InR signaling in *Tdc2*-Gal4-targeted octopaminergic neurons does not change appetitive STM in 16 hr-starved flies using 2 M as reinforcer. (**C**) Blocking InR signaling in *Tdc2*-Gal4-targeted octopaminergic neurons in *TβhnM18* mutants restored

*Figure 7 continued*

STM to control levels. (**D**) A cold shock did not disrupt emerging memory in $T\beta h^{nM18}$ mutants with blocked InR under the control of the Tdc2-Gal4 driver. Student's *t*-test was used to determine differences between two groups, and one-way ANOVA with Tukey's post hoc Honest Significant Difference (HSD) test to determine differences between three or more groups. n.s. = not significant; *p<0.05; **p<0.01, ***p<0.01. The letter 'a' marks a significant difference from random choice as determined by a one-sample sign test (p<0.05). (**E**) Model of memory regulation in $T\beta h^{nM18}$ mutants. LTM: long-term memory ; ITM: intermediate-term memory.

The online version of this article includes the following source data and figure supplement(s) for figure 7:

**Source data 1.** The raw data and sensory acuity data of *Figure 7*.

**Figure supplement 1.** The antibody against phosphorylated InR recognizes activated InR.

antibody that recognizes the phosphorylated form of the insulin receptor (InR). This region is highly conserved between humans and flies. First, we tested whether the antibody indeed recognized the activated insulin receptor. Therefore, we overexpressed the activated insulin receptor using the UAS-InR.A1325D transgene under the control of the *dTdc2*-Gal4 driver (*Figure 7—figure supplement 1*). The InR.A1325D protein variant mimics the human V938D protein variant, which is constitutively active (*Longo et al., 1992*) and the *dTdc2*-Gal4 driver targets octopaminergic reward neurons (*Busch et al., 2009*). The expression of activated InR resulted in increased immunoreactivity (*Figure 7—figure supplement 1*). The InR antibody therefore detects the activated InR. In the brain, activated InR is broadly expressed in a punctate manner and more specifically in the soma of *dTdc2*-Gal4-targeted neurons (*Figure 7A*). To uncouple energy sensing via the insulin receptor in reward neurons, we expressed UAS-InR.K1409A under the control of the *dTdc2*-Gal4 driver. The transgene encodes a dominant negative variant of the InR (InR^DN) and disrupts with InR function (*Wu et al., 2005*). In animals starved for 16 hr, the expression of InR^DN under the control of the *dTdc2*-Gal4 driver did not alter STM, but uncoupling of InR-dependent energy sensing in reward neurons in $T\beta h^{nM18}$ restored STM to normal levels (*Figure 7B and C*).

To investigate whether the improved STM also affects the emerging LTM of the $T\beta h^{nM18}$ mutants, we performed cold shock experiments in $T\beta h^{nM18}$ mutants in which InR^DN was expressed in octopaminergic reward neurons (*Figure 7D*). The 3 hr memory in controls is cold shock sensitive, but the memory in $T\beta h^{nM18}$ mutants is cold shock insensitive, supporting the idea that the mutants formed anesthesia-resistant memory.

Given that octopamine is a negative regulator of memory and that it is still absent in $T\beta h^{nM18}$ mutants with blocked insulin signaling on octopaminergic reward neurons, it is not surprising that ARM is still observed in the mutants.

The results can be described with the following model (*Figure 7E*). Insulin receptor signaling on reward neurons integrated the internal energy status into memory formation. When energy levels are high enough due to increased glycogen levels or food intake, the insulin pathway suppresses the formation of STM. In return, the system releases octopamine to suppress the formation of food-related LTM. In mildly starved $T\beta h^{nM18}$ mutants, the integration of the internal energy status into the reward neurons is still intact. In addition, the level of the internal energy supply combined with the loss of octopamine is still high enough to support the formation of protein biosynthesis-dependent LTM. In $T\beta h^{nM18}$ mutants, two functionally different memory traces are formed after training, an insulin receptor-sensitive appetitive STM and an LTM. In starved control animals, the glycogen levels are more severely depleted and octopamine is still present. This supports the formation of STM and an intermediate form of memory (ITM).

## Increased starvation results in overconsumption in $T\beta h^{nM18}$ mutants

To analyze whether the increased glycogen levels and the reduced sucrose reward of $T\beta h^{nM18}$ mutants correlate with food consumption, the energy requirements in flies under different starvation conditions were analyzed by measuring food intake (*Figure 8*). Control flies starved for shorter or longer periods consumed similar amounts of 5% sucrose. In contrast, $T\beta h^{nM18}$ mutants starved for 16 hr consumed significantly less sucrose than controls. Longer-starved mutants consumed approximately 34% more sucrose (*Figure 8B*). After longer starvation, the glycogen levels in $T\beta h^{nM18}$ mutants were still higher than those in controls (*Figure 4A*). Therefore, they overconsumed sucrose. The overconsumption was independent of the diet as they showed similar overconsumption when fed with a solution containing 5% yeast and 5% sucrose (*Figure 8B*). To investigate whether the internal energy status is also integrated

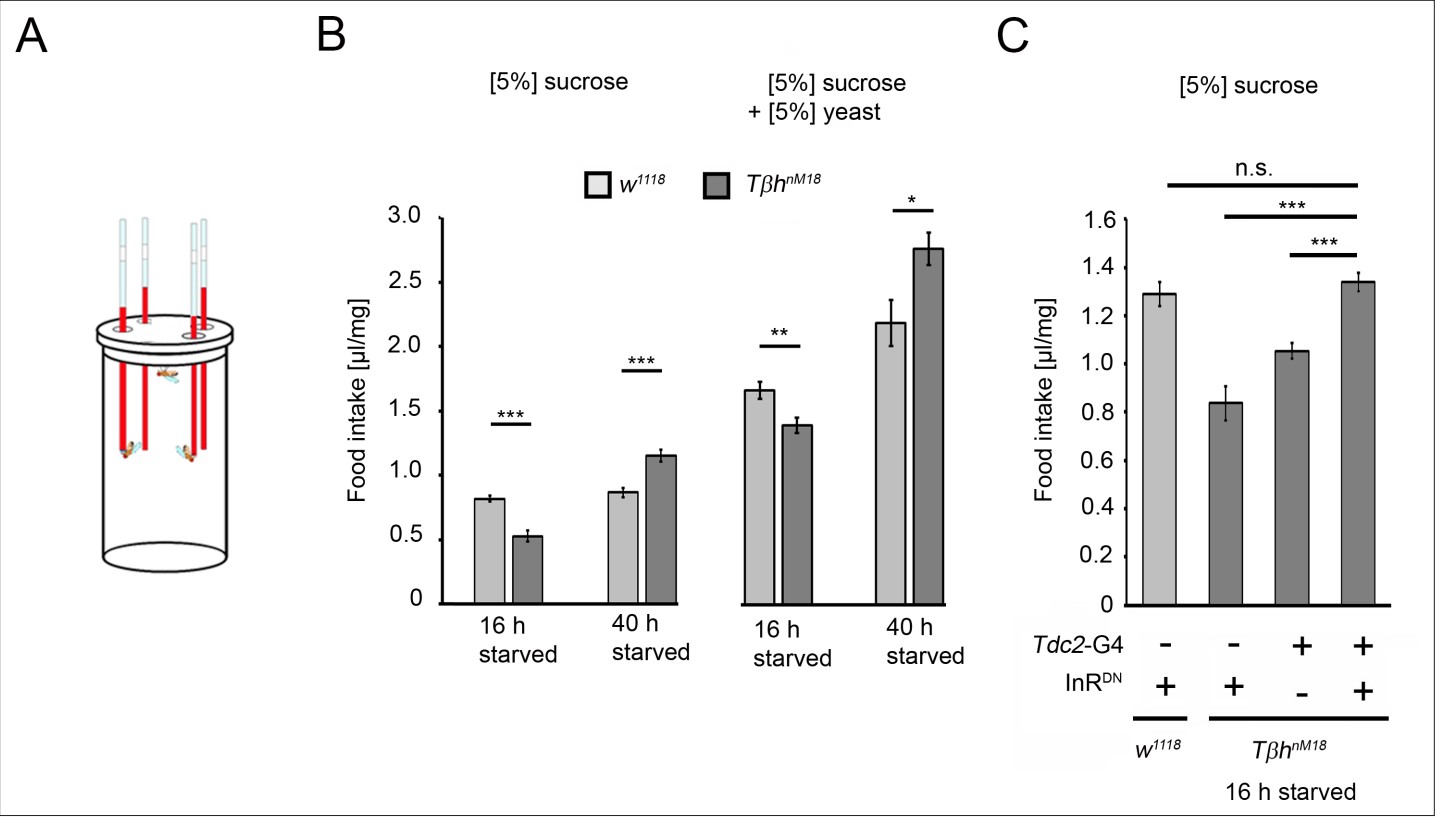

**Figure 8.** Prolonged starvation results in rebound sucrose intake in hyperglycemic $T\beta h^{nM18}$ mutants. (**A**) Capillary feeder assay used to determine food intake. (**B**) Flies were starved 16 hr or 40 hr before 24 hr food intake was measured. After 16 hr of starvation, $T\beta h^{nM18}$ mutants significantly consumed less 5% sucrose or 5% sucrose with 5% yeast. After 40 hr of starvation, $T\beta h^{nM18}$ mutants significantly consumed more sucrose and sucrose with yeast. N = 20–26 groups of eight flies. (**C**) Blocking InR signaling in *Tdc2*-Gal4-targeted octopaminergic neurons in $T\beta h^{nM18}$ mutants significantly increased 5% sucrose consumption. N = 20–28 groups of eight flies. To determine differences between two groups, Student's *t*-test was used, and to determine differences between three or more groups, one-way ANOVA with post hoc Tukey's Honest Significant Difference (HSD) was used. n.s. = non-significant; *p<0.05; **p<0.01; ***p<0.001.

The online version of this article includes the following source data for figure 8:

**Source data 1.** The raw data related of *Figure 8*.

into the feeding behavior by the octopaminergic reward neurons, we blocked insulin signaling in reward neurons in 16 hr-starved $T\beta h^{nM18}$ mutants and analyzed sucrose consumption (*Figure 8C*). The reduced sucrose consumption of $T\beta h^{nM18}$ mutants was significantly improved compared to controls. Thus, the octopaminergic reward system integrates via insulin receptor signaling the internal energy level into the regulation of food consumption. The mechanism is still intact in $T\beta h^{nM18}$ mutants.

## Discussion

Depending on the duration of starvation, flies develop STM and an anesthesia-sensitive intermediate type of a food-related memory. The extending the hunger period, memory formation shifts to an anesthesia-resistant ITM. Presentation of the reward leads to the release of octopamine and thus the suppression of LTM. The shifting of different memories phases in response to different energy supplies has been also described for memories that are formed of aversive non-food-related memories. Depending on the reduction of the internal energy supply, flies initially form protein-dependent aversive olfactory LTM and later ARM, a protein synthesis-independent form of memory (*Wu et al., 2005*). In times of energy shortage, the organism exchanges the costly protein biosynthesis-dependent LTM for the 'less costly' protein biosynthesis-independent ARM (*Mery and Kawecki, 2005*). The effect of memory transition can also be observed at the level of appetitive LTM. In $T\beta h^{nM18}$ mutants, mild starvation, along with elevated glycogen levels, leads to the formation of LTM and prolonged

starvation leads to the formation of anesthesia-resistant LTM. Furthermore, $T\beta h^{nM18}$ mutants form only anesthesia-sensitive aversive memory (*Wu et al., 2013*). In contrast, $T\beta h^{nM18}$ mutants form cold-shock-resistant LTM. Therefore, Tβh^{nM18} mutants are not defective in appetitive anesthesia-resistant types of memories rather the internal energy status defines how quickly anesthesia-resistant appetitive memories forms.

## Octopamine gates memory formation

What function does octopamine play in memory formation? Octopamine is a negative regulator of LTM as feeding octopamine receptor antagonists to control flies results in memory, and emerging memory in $T\beta h^{nM18}$ mutants is suppressed by octopamine after conditioning. Octopamine acts in the formation of LTM upstream of dopaminergic neurons as blocking neurotransmitter release of dopaminergic LTM-mediating neurons immediately after conditioning blocks memory formation in $T\beta h^{nM18}$ mutants. The results are consistent with the model describing that dopamine release is not required during the pairing of the conditioned stimulus and the reward rather than as gain control after the training (*Adel and Griffith, 2021*). The inhibitory function of octopamine controls the gain.

During the acquisition of appetitive STM, octopamine acts upstream of dopaminergic neurons M (*Burke et al., 2012*; *Liu et al., 2012*). However, at first glance, the function of octopamine appears to differ in STM and LTM. Loss of octopamine in $T\beta h^{nM18}$ mutants results in loss of STM, indicating that octopamine is required as a positive regulator for appetitive STM (*Schwaerzel et al., 2003*). The loss of STM in $T\beta h^{nM18}$ mutants has been attributed to the loss of labile STM, which forms in response to the sweetness of sucrose, but not due to defects in stable LTM, which forms in response to the caloric value of sugar (*Burke et al., 2012*; *Fujita and Tanimura, 2011*). Sweetness is sensed by taste receptor that change their sensitivity during hunger (*Inagaki et al., 2012*; *Marella et al., 2012*) and thereby influence the response to the external reinforcer. The responsiveness could be regulated by the octopaminergic system as octopaminergic neurons regulate the sucrose sensitivity of gustatory neurons in *Drosophila* (*Youn et al., 2018*). However, the reduced sensitivity of gustatory receptors cannot explain why octopamine-deficient mutants do not form STM. The $T\beta h^{nM18}$ mutants form aversive memory to lower concentrations of sucrose and the differences in sucrose preference of $T\beta h^{nM18}$ do not translate to differences in food intake during training.

The internal energy status influences how the external source of sucrose is evaluated rather than octopamine being directly involved in the acquisition of appetitive STM. Consistently, $T\beta h^{nM18}$ mutants can exhibit STM after prolonged periods of starvation. In mildly starved $T\beta h^{nM18}$ flies that do not show STM, uncoupling of energy sensing in reward neurons by blocking insulin receptor function leads to STM. Furthermore, $T\beta h^{nM18}$ mutant females with different energy requirements form appetitive STM, and a short pulse of octopamine before conditioning can even block STM. Thus, octopamine is not required in the acquisition of appetitive STM but rather suppresses LTM when enough energy is available. The function of octopamine as a suppressor of different memory types allows for selecting the memory type depending on the internal and external information. This gating function in memory formation can also be observed in other behaviors. For example, octopamine regulates the decision to approach or turn away from a food source (*Claßen and Scholz, 2018*). Taken together, the octopaminergic system integrates internal energy demands and the evaluation of external food supplies.

## The internal energy level of the animal influences memory formation

At the cellular level, insulin receptor signaling regulates energy metabolism (*Chatterjee and Perrimon, 2021*). The broad expression pattern of the insulin receptor in the *Drosophila* brain (*Figure 7*) indicates that every cell in the brain needs to regulate its own energy homeostasis. However, the fly's entire energy resources and the evaluation of the external food supply must also be integrated. The reward system evaluates the external food supply. In mice, insulin receptor signaling in dopaminergic neurons mediates food reward (*Könner et al., 2011*). Consistently, insulin receptor function in octopaminergic reward neurons regulates the rewarding properties of sucrose in appetitive STM (*Figure 6A*) and during food intake (*Figure 7C*).

The systemic metabolic rate is regulated by fat bodies. For example, after food intake, the fat body secretes unpaired 2 (Upd2), which in turn regulates the secretion of *Drosophila* insulin-like peptides via GABAergic neurons (*Rajan and Perrimon, 2012*). We show that the reduction in glycogen in the fat bodies is not sufficient to change appetitive STM, but reduction in the muscles or both tissues

is. The results support a feedback mechanism between the assessment of glycogen levels in the muscles and the brain. In addition, the muscles and the fat bodies communicate about the energy content of both structures, and this information is signaled back to the brain. Such long-range signals from the muscles exist. Skeletal muscles are secretory organs (*Pedersen and Febbraio, 2012*). For example, the muscle-secreted Amyrel amylase reduces the age-related accumulation of polyubiquitinated proteins in the brain (*Rai et al., 2021*). Regardless of how the glycogen levels of both tissues are communicated to the brain, they influence how the reinforcer is evaluated. The feedback between the energy levels in both structures is still intact in octopamine-deficient mutants as $T\beta h^{nM18}$ mutants still form STM when the energy storage is sufficiently reduced.

### Are food-related memories and internal glycogen levels predictive of food intake?

The feeding behavior in hungry animals is regulated by various neural systems, including networks that receive and process sensory information and networks that assign reward properties to food (*Berridge, 2009*). Blocking insulin receptor function in dopaminergic neurons results in increased food intake and weight gain in mice (*Könner et al., 2011*). Similarly, blocking insulin receptor function in octopaminergic reward neurons in flies with increased glycogen levels and reduced food intake increases food intake. Thus, the octopaminergic system assigns reward properties to food. However, blocking insulin receptor signaling on reward neurons does not result in extensive overconsumption, supporting that the regulation of the food quantity is still intact and is not due to insulin receptor resistance in reward neurons. Increased levels of internal energy, for example, glycogen, reduced the reward properties of food, resulting in a decreased positive association of food-related memories (*Figures 5D and 7A*) and decreased food consumption (*Figure 8*). This is intended to prevent increased food intake when there is excess food. Insulin resistance could contribute to weight gain because in addition to high internal levels of glycogen, the flies show normal food intake. However, the overconsumption only becomes visible after a prolonged period of hunger, which correlates with the emerging stable appetitive LTM. This suggests that the stability of appetitive memory could lead to the re-evaluation of food and could trigger the overconsumption of food.

In summary, the octopaminergic neurotransmitter system integrates several aspects that influence the regulation of food memories. Octopamine modulates the sensory perception of the conditioned stimulus and the sensory perception of the reward. Here, we show that the evaluation of food reward in the context of the energy storage is integrated by the octopaminergic system and influences the stability of food-related memories. The function of octopamine as a negative regulator of various forms of memories allows for selective regulation of food-related behaviors such as food intake, and the loss of this regulation could also promote dysregulation of food intake. The close relationship between the octopaminergic neurotransmitter and the noradrenergic neurotransmitter system suggests that the integrator function may be conserved.

## Materials and methods

### Key resources table

| Reagent type (species) or resource | Designation | Source or reference | Identifiers | Additional information |
|---|---|---|---|---|
| Strain, strain background (*Drosophila melanogaster*) | $w^{1118}T\beta hnM18$ | *Monastirioti et al., 1996* | | |
| Strain, strain background (*D. melanogaster*) | $w^{1118}$ | Scholz lab | | |
| Strain, strain background (*D. melanogaster*) | UAS-GlyS$^{HMS01279}$-RNAi | BDSC | BDSC_34930 | |
| Strain, strain background (*D. melanogaster*) | UAS-GlyP$^{HMS00032}$-RNAi | BDSC | BDSC_33634 | |
| Strain, strain background (*D. melanogaster*) | $w^{1118}$; UAS-shi$^{ts}$ | *Kitamoto, 2001* | | |

*Continued on next page*

*Continued*

| Reagent type (species) or resource | Designation | Source or reference | Identifiers | Additional information |
|---|---|---|---|---|
| Strain, strain background (*D. melanogaster*) | P{UAS-InR. K1409A} | BDSC | BDSC_8253 | |
| Strain, strain background (*D. melanogaster*) | *w^1118*; *FB-Gal4* | Partridge Lab | | |
| Strain, strain background (*D. melanogaster*) | *mef2-Gal4* | **Ranganayakulu et al., 1998** | | |
| Strain, strain background (*D. melanogaster*) | R15A04-Gal4 | BDSC | BDSC_48671 | |
| Strain, strain background (*D. melanogaster*) | *dTdc2-Gal4* | **Cole et al., 2005** | | |
| Chemical compound, drug | 3-Octanol | Sigma-Aldrich | 93856 | |
| Chemical compound, drug | 4-Methyl-cyclohexanol | Sigma-Aldrich | 153095 | |
| Chemical compound, drug | Paraffin oil | Sigma-Aldrich | 76235 | |
| Chemical compound, drug | Octopamin -hydrochlorid | Sigma-Aldrich | O0250-1G | |
| Chemical compound, drug | Epinastine-hydrochlorid | Sigma-Aldrich | E5156 | |
| Commercial assay or kit | Glucose (HK) assay kit | Sigma-Aldrich | GAHK20 | |
| Commercial assay or kit | Periodic Acid-Schiff (PAS) Kit | Sigma-Aldrich | 395B | |
| Antibody | Phospho-IGF-I Receptor β (Tyr1131) /Insulin Receptor β (Tyr1146) Antibody (polyclonal rabbit) | Cell Signaling Technology | 3021 | 1:50 |

## *Drosophila melanogaster*

Flies were raised on an ethanol-free standard cornmeal-molasses food at 25°C and 60% relative humidity on a 12 hr/12 hr day–night cycle. The following lines were used: *w^1118*; *w^1118*,*Tβh^nM18* (**Monastirioti et al., 1996**); *UAS-GlyS^HMS01279-RNAi* (BDSC #34930); *UAS-GlyP^HMS00032-RNAi* (BDSC #33634); *w^1118*; *UAS-shi^ts* (**Kitamoto, 2001**); *w^1118*; P{UAS-InR. K1409A} (BDSC#8253); *w^1118*;*FB-Gal4* (a generous gift from the Partridge Lab); *mef2-Gal4* (**Ranganayakulu et al., 1998**); R15A04-Gal4 (BDSC #48671); *dTdc2-Gal4* (**Cole et al., 2005**). All lines were backcrossed to *w^1118* (Scholz Lab) for at least five generations to isogenize the genetic background. For behavioral experiments, 3- to 5-day-old male flies were used, if not otherwise indicated. Male flies of the F1 generation carrying one copy of the transgene were used as controls for behavioral experiments. Animal studies using *D. melanogaster* were conducted in agreement with the regulations of the DFG and the Land North Rhine-Westphalia. Other ethics approval and informed consent statement are not applicable for research using *Drosophila*.

## Olfactory learning and memory

Associative olfactory learning and memory was trained and tested with a modified version of the Tully and Quinn olfactory conditioning apparatus (**Schwaerzel et al., 2003**; **Tully and Quinn, 1985**). Approximately 70 1- to 2-day-old male flies were collected with $CO_2$ anesthesia and were kept for 2 d at 25°C to recover from $CO_2$ sedation. Briefly, 3- to 5-day-old male flies were starved for 16 hr or 40 hr in vials with water-soaked filter paper at the bottom. Flies were transferred to training tubes and exposed to the first odorant for 2 min, either 3-octanol (3-OCT diluted 1:80 in paraffin oil) or 4-methylcyclohexanol (MCH diluted 1:100 in paraffin oil). Then, they were transferred to a second tube and exposed to the second odorant in the presence of filter paper soaked with either 0.15 M sucrose, 2 M sucrose, or 5% yeast. To analyze the abilities of the flies to learn and remember the odorant paired with the reward, 2 min, 3 hr, 6 hr, or 24 hr after training, flies were given the choice between odorant 1 (CS+) and odorant 2 (CS-) for 2 min. The performance index (PI) was calculated as PI = (# (CS+) + # (CS-)/(total # flies)), where CS+ indicates the odorant associated with the appetitive reinforcer and CS- indicates the odorant that was not associated with the reinforcer. To exclude

non-associative effects, each 'n' consists of a reciprocally trained, independent group of naïve flies. For pharmacological experiments, flies were fed water, 3 mM OA or 2 M sucrose before training or with 3 mM OA or 3 mM epinastine between training and testing. The assay was performed at room temperature (RT) with 60% relative humidity. For cold shock experiments, flies in vials were placed in ice-cold water for 2 min.

Odorant acuity and odorant balance were tested with naïve flies that were not used later for learning and memory experiments. For testing, the flies were placed into the Tully Quinn setup. They chose between the odorant- or the solvent paraffin oil-containing sides or between both odorants for 2 min. The sucrose and yeast preference was determined by analyzing whether they preferred the side with a sucrose or yeast-soaked filter paper over a water-soaked filter paper. Flies of all genotypes used for behavioral experiments were tested for odorant acuity, odorant balance, and ability to sense reinforcer. To determine sucrose intake in the Tully and Quinn olfactory conditioning apparatus, flies were fed with a food-safe-colored sucrose solution and food intake was analyzed by the color in the abdomen. The data are provided in the supplement to the respective figure.

## Food intake

To analyze the consumption of nutrients in flies, the capillary feeder (CaFe) assay was performed (*Diegelmann et al., 2017*; *Ja et al., 2007*). Briefly, eight 3- to 5-day-old male or female flies that were either non-starved or starved for 18 hr or 40 hr had access to four capillaries filled with either 5% sucrose or 5% yeast or a mixture of both for 24 hr at 25°C and 60% humidity. The solution was colored with red food dye. During starvation, flies were kept in vials with wet filter paper at 25°C and 60% humidity. The amount of consumed solution was determined using an electronic caliper. To account for evaporation, the average evaporation of three CAFE setups without flies was measured and used to normalize the average food intake per fly. To normalize food intake to body weight, at least five times 100 male flies per genotype and condition were weighed, and the average body weight of a single fly was determined. The total consumption per µg fly was calculated by dividing the total consumption per fly by the mean weight of a fly. N indicates the number of tested groups.

## Glycogen content

Whole-body glycogen levels were determined with the Glucose (HK) Assay Kit (Sigma-Aldrich, #GAHK20-1KT) according to the protocol of *Tennessen et al., 2014*. A group of five male flies that were either sated or starved for 16 hr or 40 hr were homogenized in 100 µl ice-cold 1× PBS. To reduce enzymatic degradation, proteins and enzymes were heat inactivated at 70°C for 10 min. The supernatant was removed and diluted 1:3 with 1× PBS. Then, 20 µl of the samples were either added to 20 µl of 1× PBS or PBS/amyloglucosidase mix and incubated at 37°C for 60 min. Then, 100 µl of HK-reagent was added to the sample or glucose standard and incubated for 15 min at RT. Absorbance was measured at 340 nm. The glycogen content was calculated by subtracting the total glucose concentration from 1× PBS-treated samples from the total glucose of amyloglucosidase-treated samples.

## Periodic acid staining

To visualize glycogen levels in the fat bodies and muscle tissue of larvae and adult *Drosophila*, PAS staining was performed after *Yamada et al., 2018* with a slight modification. The samples were fixed in 3.5% formaldehyde for 20 min and washed two times for 5 min with 1% BSA/PBS. Periodic acid solution was added for 5 min, followed by two washes of 5 min with 1% BSA/PBS. Schiff's reagent was added for 5 min, followed by two washes of 5 min with 1% BSA/PBS. Tissue was stored in 50% glycerol.

## Immunohistochemistry

For immunohistochemistry, antibodies raised in rabbits against the activated insulin-like receptor (Cell Signaling Technology #3021) were used at a dilution of 1:50 in 5% normal goat serum in PBS with 0.1% Triton and incubated for 2 d. The brains were washed with PBS with 0.3% Triton.

## Quantification and statistical analysis

Food intake was displayed as the mean ± s.e.m. For learning and memory experiments, the data are displayed as boxplot ± minimum (Q1 – 1.5 * IQR) and maximum (Q1 + 1.5*IQR). We used the

Shapiro–Wilk test (significance level p<0.05) followed by a QQ-Plot chart analysis to determine whether the data were normally distributed. For normally distributed data, we used the Student's *t*-test to compare differences between two groups and the one-way ANOVA with Tukey's post hoc Honest Significant Difference (HSD) test for differences between more than two groups. For nonparametric data, we used the Mann–Whitney *U* test for differences between two groups and for more than two groups the Kruskal–Wallis test with post hoc Dunn analysis and Bonferroni correction. The nonparametric one-sample sign test was used to analyze whether behavior was not based on random choice and differed from zero (p<0.05). The statistical data analysis was performed using statskingdom (https://www.statskingdom.com). Boxplots were generated with Microsoft Excel 2016 and GIMP 2.10.12.

## Acknowledgements

We thank the Linda Partridge lab and the Bloomington Drosophila Stock Center for providing *Drosophila* fly lines. HS was supported by SFB1340, DFG-Scho656-10-1, and the Hetzler award.

## Additional information

### Funding

| Funder | Grant reference number | Author |
| --- | --- | --- |
| Deutsche Forschungsgemeinschaft | Scho 656/10-1 | Henrike Scholz |
| Bloomington Drosophila Stock Center | SFB1340 | Henrike Scholz |

The funders had no role in study design, data collection and interpretation, or the decision to submit the work for publication.

### Author contributions

Michael Berger, Conceptualization, Formal analysis, Validation, Investigation, Visualization, Writing - original draft; Michèle Fraatz, Validation, Investigation; Katrin Auweiler, Tanna El Khadrawe, Investigation; Katharina Dorn, Formal analysis, Validation, Investigation; Henrike Scholz, Conceptualization, Resources, Data curation, Formal analysis, Supervision, Funding acquisition, Validation, Visualization, Writing - original draft, Project administration, Writing - review and editing

### Author ORCIDs

Katrin Auweiler http://orcid.org/0009-0000-8598-4616
Henrike Scholz http://orcid.org/0000-0001-8619-5328

Reviewer #1 (Public Review): https://doi.org/10.7554/eLife.88247.3.sa1
Reviewer #2 (Public Review): https://doi.org/10.7554/eLife.88247.3.sa2
Reviewer #3 (Public Review): https://doi.org/10.7554/eLife.88247.3.sa3
Author response https://doi.org/10.7554/eLife.88247.3.sa4

## Additional files

### Supplementary files
• MDAR checklist

### Data availability

All data generated or analysed during this study are included in the manuscript and supporting files. All data related to figures are included in the *Figure 1—source data 1* to *Figure 8—source data 1*.

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
