## [Editor Report · eLife assessment]

This **important** study dissects the role of octopamine in the interplay between internal energy homeostasis, food intake, and food-related memories. The **solid** experimental evidence will shed additional light on previously published work and should be of interest to the growing community of biologists interested in how internal state shapes behavior, including decision-making processes, learning and memory.

---

## [Referee Report · Reviewer #1 (Public Review)]

The present study conducted by Berger et al. delves into the extent to which memory formation relies on available energy reserves. While aversive memory formation has been extensively studied in this context, the investigation into appetitive memory formation has been comparatively sparse. It has long been recognized that flies can only form appetitive memory under conditions of starvation. However, the authors of this study go beyond this understanding by revealing that not only the duration of starvation matters, but it also dictates the type of memory formed, whether short- or long-term memory. The authors illustrate that internal glycogen stores play a crucial role in this process, facilitated by insulin-like signaling in octopaminergic reward neurons, which integrates internal energy reserves into memory formation. Consequently, the authors propose that octopamine serves as a negative regulator of various forms of memory, shedding light on the enduring question of the octopaminergic neuronal system's involvement in appetitive memory formation, which has been overshadowed by the focus on the dopaminergic system in recent years. Additionally, the findings contribute to the ongoing debate concerning the role of insulin receptors, whether they function within neurons themselves or in glial cells. Moreover, the authors not only convincingly demonstrate that octopamine negatively regulates appetitive memory formation, but they also propose a mechanism whereby the insulin receptor in octopaminergic neurons senses the internal energy status and subsequently modulates the activity of these neurons. The experiments are meticulously designed, employing a variety of behavioral assays, genetic tools for manipulating neuronal activity, and state-of-the-art imaging techniques. The conclusions are well supported by the data and carefully performed controlled experiments, yielding high-quality data.

---

## [Referee Report · Reviewer #2 (Public Review)]

How organism physiological state modulates establishment and perdurance of memories is a timely question that the authors aimed at addressing by studying the interplay between energy homeostasis and food-related conditioning in *Drosophila*. Specifically, they studied how starvation modulates the establishment of short-term vs long-term memories and clarified the role of the monoamine Octopamine in food-related conditioning, showing that it is not per se involved in formation of appetitive short-term memories but rather gates memory formation by suppressing LTM when energy levels are high. This work clarifies previously described phenotypes and provides insight about interconnections between energy levels, feeding and formation of short-term and long-term food-related memories.

Strengths

- Previous studies have documented the impact of Octopamine on different aspects of food-related behaviors (regulation of energy homeostasis, feeding, sugar sensing, appetitive memory...), but we currently lack a clear understanding of how these different functions are interconnected. The authors have used a variety of experimental approaches to systematically test the impact of internal energy levels in establishment of different forms of appetitive memory and the role of Octopamine in this process.

- The authors have used a range of approaches, performed carefully controlled experiments and produced high quality data.

---

## [Referee Report · Reviewer #3 (Public Review)]

In this manuscript, Berger et al. study how internal energy storage influence learning and memory. Since in *Drosophila melanogaster*, octopamine (OA) is involved in the regulation of energy homeostasis they focus on the roles of OA. To do so they use the tyramine-β-hydroxylase (Tbh) mutant that is lacking the neurotransmitter OA and study short term memory (STM), long-term memory (LTM) and anesthesia-resistant memory (ARM). They show that the duration of starvation affects the magnitude of both short- and long-term memory. In addition, they show that OA has a suppressive effect on learning and memory. In terms of energy storage, they show that internal glycogen storage influences how long sucrose is remembered, and high glycogen suppresses memory. Finally, they show that insulin-like signaling in octopaminergic neurons, which is also related to internal energy storage, suppresses learning and memory.

The revised version of the manuscript is greatly improved, and I thank the authors for taking the comment seriously. This is an important study that extends our knowledge on OA activity in learning and memory and the effects the metabolic state has on learning and memory. The authors nicely use the genetic tools available in flies to try and unravel the complex circuitry of metabolic state level, OA activity and learning and memory. The overall take-home message of the manuscript is clear and supported by the data presented.

---

## [Author Response]

The following is the authors’ response to the original reviews.

**Reviewer #1 (Public Review):**
The present study by Berger et al. analyzes to what extent memory formation is dependent on available energy reserves. This has been dealt with extensively in the case of aversive memory formation, but only very sparsely in the case of appetitive memory formation. It has long been known that an appetitive memory in flies can only be formed by starvation. However, the authors here additionally show that not only the duration of starvation plays a role, but also determines which form of memory (short- or long-term memory) is formed. The authors demonstrated that internal glycogen stores play a role in this process and that this is achieved through insulin-like signaling in octopaminergic reward neurons that integrates internal energy stores into memory formation. Here, the authors suggest that octopamine plays a role as a negative regulator of different forms of memory.The study sheds light on an old question, to what extent the octopaminergic neuronal system plays a role in the formation of appetitive memory, since in recent years only the dopaminergic system has been in focus. Furthermore, the data are an interesting contribution to the ongoing debate whether insulin receptors play a role in neurons themselves or in glial cells. The experiments are very well designed and the authors used a variety of behavioural experiments, genetic tools to manipulate neuronal activity and state-of-the-art imaging techniques. In addition, they not only clearly demonstrated that octopamine is a negative regulator of appetitive memory formation, but also proposed a mechanism by which the insulin receptor in octopaminergic neurons senses the internal energy status and then controls the activity of those neurons. The conclusions are mostly supported by the data, but some aspects related to the experimental design, some explanations and literature references need more clarification and revision.(1) Usually, long-term memory (LTM) is tested 24 hours after training. Here, the authors usually refer to LTM as a memory that is tested 6 hours after training. The addition of a control experiment to show that LTM that the authors observe here lasts longer would increase the power of this study immensely.

We thank the reviewer for this comment, as it helped greatly to clarify the matter.

We measured memory of control and mutant flies 24 h after the training and included the data into the manuscript (Figure 1B and summarized in a model in Figure 2C). We show that control flies develop an intermediate type of memory, that is depending on the length of starvation either anesthesia-sensitive or resistant. Mutants lacking octopamine develop either anesthesia-sensitive or resistant long-term memory.

(2) The authors define here another consolidated memory component as ARM, when they applied a cold-shock 2 hours after training. However, some publications showed that LTM is formed after only one training cycle (Krashes et al 2008, Tempel et al 1983). This makes it difficult to determine, whether appetitive ARM can be formed. Furthermore, one study showed that appetitive ARM is absent after massed training (Colomb et al 2009). Therefore, the conclusion could be also, that different starvation protocols, would lead to different stabilities of LTM. Therefore, additional experiments could help to clarify this opposing explanation. From these results, it can then be concluded either that different stable forms of LTM are formed depending on the starvation state, or that two differently consolidated memory phases (LTM, ARM) are formed, as has already been shown for aversive memory. This is also important for other statements in the manuscript, and therefore the authors should address this. For example, the findings about the insulin receptor (is it two opposing memories or different stabilities of LTM).

The flies indeed develop different types of memory depending on the length of starvation and the internal energy supply.

**Reviewer #2 (Public Review):**
How organism physiological state modulates establishment and perdurance of memories is a timely question that the authors aimed at addressing by studying the interplay between energy homeostasis and food-related conditioning in *Drosophila*. Specifically, they studied how starvation modulates the establishment of short-term vs long-term memories and clarified the role of the monoamine Octopamine in food-related conditioning, showing that it is not per se involved in formation of appetitive short-term memories but rather gates memory formation by suppressing LTM when energy levels are high. This work clarifies previously described phenotypes and provides insight about interconnections between energy levels, feeding and formation of short-term and long-term food-related memories. In the absence of population-specific manipulation of octopamine signaling, it however does not reach a circuit-level understanding of how these different processes are integrated.StrengthsPrevious studies have documented the impact of Octopamine on different aspects of food-related behaviors (regulation of energy homeostasis, feeding, sugar sensing, appetitive memory...), but we currently lack a clear understanding of how these different functions are interconnected. The authors have used a variety of experimental approaches to systematically test the impact of internal energy levels in establishment of appetitive memory and the role of Octopamine in this process.The authors have used a range of approaches, performed carefully controlled experiments and produced high quality data.Weaknesses(1) In the tbh mutant flies, Tyramine -to- Octopamine conversion is inhibited, resulting not only in a lack of Octopamine, but also in elevated levels of Tyramine. If and how elevated levels of Tyramine contributes to the described phenotypes is unclear. In the current version of the manuscript, only one set of experiments (Figure 2) has been performed using Octopamine agonist. This is particularly important in light of recent published data showing that starvation modifies Tyramine levels.(2) Octopamine (and its precursor Tyramine) have been implicated in numerous processes, complicating the analysis of the phenotypes resulting from a general inhibition of tbh.

We thank the reviewer for raising these points. The observed memory defects of the Tbh mutants can be solely explained by loss of octopamine. We included models into the manuscript to illustrate this (Figure 2 C and Figure 7E).

To address whether the elevated levels of tyramine observed in Tbh mutants interfere with food consumption, we analyzed the effect of increased levels of tyramine and octopamine on food consumption. We included the data (Figure S2). An increase in tyramine levels did not result in a change in food intake, rather the increase in octopamine levels reduced food intake. Our data show that the reduction of food intake observed in starved Tbh mutants is due to the increased internal energy supply.

(3) The manuscript explores various aspects of the impact of energy levels on food-related behaviors and the underlying sensing and effector mechanism, both in wild-type and tbh mutants, making it difficult to follow the flow of the results.

We included models illustrating the results to clarify the content of the manuscript.

**Reviewer #3 (Public Review):**
In this manuscript, Berger et al. study how internal energy storage influence learning and memory. Since in *Drosophila melanogaster*, octopamine (OA) is involved in the regulation of energy homeostasis they focus on the roles of OA. To do so they use the tyramine-β-hydroxylase (Tbh) mutant that is lacking the neurotransmitter OA and study short term memory (STM), long-term memory (LTM) and anesthesia-resistant memory (ARM). They show that the duration of starvation affects the magnitude of both short- and long-term memory. In addition, they show that OA has a suppressive effect on learning and memory. In terms of energy storage, they show that internal glycogen storage influences how long sucrose is remembered and high glycogen suppresses memory. Finally, they show that insulin-like signaling in octopaminergic neurons, which is also related to internal energy storage, suppresses learning and memory.This is an important study that extends our knowledge on OA activity in learning and memory and the effects the metabolic state has on learning and memory. The authors nicely use the genetic tools available in flies to try and unravel the complex circuitry of metabolic state level, OA activity and learning and memory.Nevertheless, I do have some comments that I think require attention:(1) The authors use RNAi to reduce the level of glycogen synthase or glycogen phosphorylase. These manipulations are expected to affect the level of glycogen. Using specific drivers the authors attempt to manipulate glycogen level at the muscles and fat bodies and examine how this affects learning and memory. The conclusions of the authors arise solely from the manipulation intended (i.e. the genetics). However, the authors also directly measured glycogen levels at these organs and those do not follow the manipulation intended, i.e. the RNAi had very limited effect on the glycogen level. Nevertheless, these results are ignored.

We agreed with the reviewer and repeated the experiments. While we could not detect differences in whole animals, we detected differences in tissues enriched for muscles or fat, e.g. thorax or abdomen. We added the data.

(2) The authors claim in the summary that OA is not required for STM. However, according to one experiment OA is required for STM as Tbh mutants cannot form STM. In another experiment OA is suppressive to STM as wt flies fed with OA cannot form STM. Therefore, it is very difficult to appreciate the actual role of OA on STM.

During mild starvation, the internal energy supply is greater in Tbh mutants than in control flies. This information is integrated into the reward system via insulin receptor signaling. Therefore, the association between the odorant and sucrose is not meaningful to the mutants and no STM is formed. At the same time there is no release of octopamine and therefore no repression of LTM. In starved animals, octopamine suppresses food intake (we added the data). This is consistent with a function of Octopamine as a signal for the presence of food. Depending on when the signal comes, this might suppress the formation of STM or LTM.

(3) The authors use t-test and ANOVA for most of the statistics, however, they did not perform normality tests. While I am quite sure that most datasets will pass normality test, nevertheless, this is required.

Thanks for pointing this out. We have included a description in the “Materials and Methods” section that explains how we tested the data for normal distribution. We corrected the figure legends accordingly.

“We used the Shapiro-Wilk test (significance level P < 0.05) followed by a QQ-Plot chart analysis to determine whether the data were normal distributed. “

(4) While it is logical to assume that OA neurons are upstream to R15A04 DA neurons, I am not sure this really arises from the experiment that is presented here. It is well established that without activity in R15A04 DA neurons there is no LTM. Since OA acts to decrease LTM, can one really conclude anything about the location of OA effect when there is no learning?

Normally control flies did not form memory 6 h after training, only Tbh mutants. We wanted to investigate what kind of memory develops in Tbh mutants. During the experiments of the manuscript, we kept the training procedure constant.

(5) It is unclear how expression of a dominant negative form of insulin receptor (InR) in OA neurons can rescue the lack of OA due to the Tbh mutation. If OA neurons cannot release anything to the presumably downstream DA neurons, how can changing their internal signaling has any effect?The expression of the dominant negative form of the insulin receptor signals no food or low energy levels and activation of the insulin receptor that there is enough food. The reward is a source of food, but the energy content is not high enough to fill the energy stores. The insulin receptor activation can activate at least three different signaling cascades, one of which might regulate octopamine release.While I stressed some comments that need to be addressed, the overall take-home message of the manuscript is supported and the authors do show that the metabolic state of the animal affects learning and memory. I do think though, that some more caution is required for some of the conclusions.

We added additional data to address the points raised.

**Recommendations for the authors:**

We addressed all points raised by the reviewers, clarified the content or added more data.

**Reviewer #1 (Recommendations For The Authors):**
(1) Throughout the manuscript, the full stop of a sentence is always placed before the references.

We fixed this.

(2) I find the English in the manuscript not yet sufficient for publication. I suggest that the authors carefully revise the manuscript. I think if the sentences are structured a little more clearly, this paper has enormous potential to be read by your broad community.

We agree and revised the manuscript. We hope the manuscript is now clearer.

(3) Sentences l114 to l117 are misleading. The authors imply that they tested the same flies for changes in odor perception or sucrose sensitivity. I assume that the authors meant that they analyzed different groups of animals.

We clarified the sentence as follows:

“To ensure that the observed differences in learning and memory were not due to changes in odorant perception, odorant evaluation or sucrose sensitivity, different fly populations of the same genotypes were tested for their odorant acuity, odorant preference and their sucrose responsiveness (Table S1).”

(4) In the title as well as in the abstract the influence of octopamine on appetitive memory formation is described in more detail, this is also the main focus of this study. However, in the introduction, the influence of the insulin receptor on memory formation is discussed first. Personally, I would describe this later in the manuscript, ideally in the results section. At this point in the manuscript, this leads to an interruption in the flow of reading.

Thanks for the suggestion. We changed the order in the introduction.

(5) The authors could consider, since they only used *Drosophila melanogaster*, changing "*Drosophila melanogaster*" to "*Drosophila*" throughout the manuscript.

We modified the text accordingly.

(6) All evaluations and statistical tests are state of the art. However, I have one comment. For each statistical test, a correction should be made depending on the number of tests. However, I could not determine whether this was also done for the parametric or non-parametric one-sample t-test. From the results and the methods section, I would guess not. Here I would recommend a Bonferroni correction or even better a Sidak-Holm correction. Furthermore, the authors could also go into more detail about which non-parametric one-sample t-test they used.

We described the statistic used in more detail in the material and method section.

“We used the Shapiro-Wilk test (significance level P < 0.05) followed by a QQ-Plot chart analysis to determine whether the data were normal distributed. For normal distributed data, we used the Student’s t test to compare differences between two groups and the one-way ANOVA with Tukey’s post hoc HSD test for differences between more than two groups. For nonparametric data, we used the Mann-Whitney U test for differences between two groups and for more than two groups the Kruskal-Wallis test with post hoc Duenn analysis and Bonferroni correction. The nonparametric one-sample sign test was used to analyze whether behavior was not based on random choice and differed from zero (P < 0.5). The statistical data analysis was performed using statskingdom (https://www.statskingdom.com).”

(7) In nearly all figure legends the sentence "The letter "a" marks a significant difference from random choice as determined by a one-sample sign test (P* < 0.05; P*< 0.01)" occur. This is correctly indexed in the figures. However, I do not understand here what then P < 0.05; P**< 0.01 means. The significance level should be described here. I would strongly recommend the authors to make the definition clearer.

We corrected this in the figure legends (see also above).

(8) In Fig. 1B the labelling is a bit confusing. I interpret the two right groups as the mutants for octopamine, but there is still w[1118] in front.

We modified the Figure 1B.

**Reviewer #2 (Recommendations For The Authors):**
Suggestions(1) Assessing the contribution of Tyramine in the observed phenotypes (for example by reducing the levels of Tyramine or its specific receptor) would help understand the contribution of Tyramine in the observed phenotypes.See comments above.

We thank the reviewer for raising these points. The observed memory defects of the Tbh mutants can be solely explained by loss of octopamine. We included models into the manuscript to illustrate this (Figure 2 C and Figure 7E).

To address whether the elevated levels of tyramine observed in Tbh mutants interfere with food consumption, we analyzed the effect of increased levels of tyramine and octopamine on food consumption. We included the data (Figure S2). An increase in tyramine levels did not result in a change in food intake, rather the increased octopamine levels reduced food intake. Our data show that the reduction of food intake observed in starved Tbh mutants is due to the increased internal energy supply.

(2) Cell-specific inhibition of octopamine receptors should thus be performed to precisely interpret the observed phenotypes and dissect how interconnected the different phenotypes are, which is the object of this publication.

We observed that the time point and duration of octopamine application changes the behavioral output. The behavior analyzed depends on pulses of octopamine and differences of the internal energy status. A cell-specific inhibition via RNAi knock down of octopamine receptors might not clarify the issue.

(3) Defining of streamline and progressively integrating the different observations into a unifying model would improve the clarity and flow of the manuscript.

We included models explaining the observed results (Figure 2C and Figure 7E).

Minor commentsLine 129: Figure 1B should be mentioned, not 2B.Figure 1 legend: E should be replaced by C (after A,B).Figure S5: what are the arrows pointing to? Why are the Inr foci visible in A not seen in B? It should be mCD8-GFP and not mCD on top of the images.

We fixed this.

**Reviewer #3 (Recommendations For The Authors):**
Major:(1) Can one really conclude from Figure 2A that OA acts on R15A04 DA neurons? It is well established that without activity in these DA neurons there is no LTM. Since OA acts to decrease learning, how one can conclude anything about the location of OA effect when there is no learning? With STM the situation was opposite, OA supported learning and this was abolished when DA neurons were silenced.I think some supporting experiment are required, i.e. how OA affects DA neurons activity or, alternatively, tone down a bit the writing.

Normally control flies did not form memory 6 h after training, only Tbh mutants. We wanted to investigate what kind of memory develops in Tbh mutants. During the experiments of the manuscript, we kept the training procedure constant. The inhibition of dopaminergic neurons blocks the memory of Tbh mutants. Taken together the duration of the memory, the cold-shock experiments and the inhibition of the dopaminergic neurons, Tbh develops LTM after training. This training does not evoke memory in controls.

The loss of STM in mildly starved Tbh mutants depends on the integration of the high internal energy levels via InR signaling. Reducing the internal energy levels further by extension of starvation result in STM supporting that OA is not directly involved in the formation of STM.

(2) Figure 4 requires some clarifications. In Supplementary Figure S2 the authors show that they could not manipulate glycogen levels in muscles. However, in Figure 4B they show that "Increasing glycogen levels in the muscles did not change short-term memory in 16 h starved flies, but the reduction in glycogen significantly improved memory strength (Figure 4B)" (lines 231-233). How can this be reconciled?

While we could not detect differences in whole animals, we detected differences in glycogen content in body parts enriched with muscles or fat, e.g. thorax or abdomen when using UAS-GlyP-RNAi or UAS-GlyS-RNAi under the control of the respective Gal4 drivers.

We added the data.

Likewise, the authors write that "Increasing or decreasing glycogen levels in the fat bodies had no effect on memory performance (Figure 4C)" Line (233-234). However, in Figure S2 they show that they can only increase glycogen levels but not decrease them.As explained above the conclusion of Figure 4 "Thus, low levels of glycogen in the muscles upon starvation positively influence appetitive short-term memory, while high levels of glycogen in the muscles and fat body reduce short-term memory" lines 245-246, is not supported by the direct measurements of glycogen presented in Figure S2.

We added the data showing that the reduction or increase can be measured when analyzing the specific body parts enriched in muscles tissue or fat tissue.

(3) In cases where mutant flies do not display learning, a control should be done to see if they ate the sugar (with dye). Especially since the genetic manipulation affects metabolism.

We analyzed how much sucrose the animals consumed in the behavioral test. Tbh and controls fed and there was no difference in feeding behavior between the mutants and the controls.

“We next determined whether differences in preferences influence sucrose intake during training. Therefore, we measured the sucrose intake of starved flies in the behavioral set up. We used a food-colored sucrose solution and evaluated the presence of food in the abdomen of the fly after two 2 min (Table S1). Flies fed sucrose within 2 min and there was no difference between w1118 and TβhnM18 flies. “

(4) The use of t-test requires the data to be normally distributed. If I am not mistaken this was not demonstrated for any of the datasets used. I did a quick check on one of the datasets provided in the excel sheet and it is normally distributed. Therefore, please add normality test for all data sets. If some do not pass normality, please use a suitable non-parametric test.

We added normality test to all data sets and used non-parametric tests for non-normal distributed data. We clarify this in the material and method section and the figure legends.

(5) The authors show that OA suppresses also STM. This result is in contradiction to previous published results. This by itself is not a problem. However, this result also seems to me in contradiction to the authors own results. According to Figure 1B, OA is required for STM as it absence in the tbh mutant results in loss of STM. According to Figure 2C, OA is reducing STM as wt flies fed with OA just prior to learning do not form STM. This appears in other places in the manuscript as well.In addition, in the text lines 178-180, the authors write "A short pulse of octopamine before the training inhibits the STM. Thus, octopamine is a negative regulator of appetitive dopaminergic neuron-dependent long-term memory and can block STM." But in the summary they write "Octopamine is not required for short-term memory, since octopamine deficient mutants form appetitive short-term memory to sucrose and to other nutrients depending on the internal energy status."So, the take-home message regarding OA and STM is unclear.The authors need to better clarify this point.

We clarified these points. See comments above. The loss of memory in Tbh mutants is not due to loss of octopamine, but increased energy levels that changes the reward properties of sucrose.

(6) The manuscript is very difficult to follow. The authors constantly change between 16 and 40 hours starvation, short term memory, 3 hour memory and 6 hour memory. I think it would have been better to have a more focused manuscript. However, if this is not possible, I recommend preparing a diagram with the different neurons or signaling pathways (i.e. insulin) and how they affect each other. Also, perhaps add to each figure a panel describing exactly the experimental conditions. I think also simplifying the text and adding more conclusions throughout the results section will help the readers to follow. Finally, I think that it would help understanding the conclusions if the authors can add a diagram of the flow that they think occurs. For example, the authors show that glycogen suppresses learning as its reduction increases learning. They also show that InR activity receptor suppresses learning as its KD also increases learning. If I am not mistaken the link between the two is not straight forward (but I may be wrong here). A diagram of the flow would be very helpful.

We prepared diagrams summarizing and explaining the results.

Minor(1) I may not have understood correctly as I am not sure that I found Table S1.Also, there was no legend for Table S1.Nevertheless, if I understood correctly, the authors write that "Before the experiments, flies were tested to determine whether they perceived the odorants, preferred one odorant over other and responded to the reward similarly to ensure that the observed differences in behavior were not due to changes in odorant perception or sucrose sensitivity (Table S1)." However, according to the Table that I found it seems that following 40h starvation wt flies show preference to OCT whereas this does not occur for the mutant. Also, it seems that at 16h the mutant has a much higher preference to the odors than after 40h. This is a bit odd. I am also not sure what the balance value refers to. Finally, the mutant shows really low 2M sucrose preference after 40h. In general, this set of experiments requires a bit more explanation.I think it is better to show these experiments using graphs and add this to the supplementary figures.

We clarified the experiments in the result section as follows and added an explanation to the material and method section. We tested the odorant acuity and sucrose preference for all genotypes used in the manuscript and added the data to the Table S1.

“The flies of the different genotypes sensed the odorants and evaluated them as similar salient in comparison. This is important to a avoid a bias in the situation where flies have to choose between the two odorants after training. They also sensed sucrose. We next determined whether differences in preferences influence sucrose intake during training. Therefore, we measured the sucrose intake of starved flies in the behavioral set up. We used a food-colored sucrose solution and evaluated the presence of food in the abdomen of the fly after two 2 min (Table S1). Flies fed sucrose within 2 min and there was no difference between w1118 and TβhnM18 flies.”

(2) Line 129 should be Figure 1B

Is corrected.

(3) Line 133, Figure 1C, how can one explain the negative reinforcement? I can understand no reinforcement, but negative?

The effect of glucose might be doses dependent. 0.15 M sucrose is a much closer to a realistic concentration found in fruits than 2 M sucrose and might therefore elicit aversion. When animals are starved enough they might find any food source attractive, even when the concentrations of sucrose is unrealistic.

(4) Figure 1, why are the graphs different between panel B and C?

Is corrected.

(5) In Figure S1, are the TβhnM18 groups differ significantly from zero? I think they are, so better to state this somewhere. If not, the claims in lines 134-135 are not supported by the data.

We added the significance and added the data to Figure 1.

Figure S1 legend: there is no A panel. Also "below box blots" should be box plots.

Thanks for pointing that out. We corrected it.

(6) It is not clear what is the duration of starvation used in Figure 2A. I assume that 16h and sucrose 2M used were used, but I would state that explicitly.

We added the information to the figure legends.

(7) Figure 2A is missing a control of flies with both the driver and UAS shibirets at the permissive temperature.

We added the controls to the supplement (Figure S1).

(8) It seems to me that Figure 3B, in which the author state that "Only after 40 h of starvation did TβhnM18 mutants show a similar preference to control sucrose consumption" (line 198) is somewhat in contradiction to Table S1 in which I see Sucrose preference for wt 0.36 and for tbh 0.17. I think this comment arise because I did not understand Table S1 correctly, so please better explain.

We rewrote this section.

(9) In Figure 3C, consider not using std as this stands for standard deviation and may be confusing.

We now use the term “food” instead of “std” and explained in the legend that food means standard fly food.

We fixed this.

(10) Please check the Supplementary Figures. I think Figures S2 and S3 are switched.

We fixed this.

(11) There is a mistake in Figure S3A. The right column should have another "+" sign.

Thanks, we fixed this.

(12) I am somewhat puzzled by Figures 4 and 5. If I understand correctly figure 4B w1118 mef2-G4 is exactly the same experiment as Figure 5A w1118 mef2-G4 and yet in Figure 4B performance index is 0.2 and in Figure 5A about 0.4. According to other comparisons it seems to me that these will be significantly different and yet it is the same experiment.

They are two independent experiments done at different times. The controls were independently repeated.

(13) Line 273 should be Figure 5C.

Is corrected.

(14) I don't think this is a correct sentence "Virgin females remembered sucrose significantly better than mated females." Line 274.

Reads now:

“Virgin females remembered the odorant paired with sucrose significantly better than mated females.”

(15) Line 340 there is no Figure 1E

Is fixed (1 C)

(16) The data excel file is difficult to follow. In Figure 2 there are references to Figure 5. The graphs are pointing to other files. Text is not always in English. It is not clear what W stands for. I recommend making it more accessible.

We corrected the data excel files.

(17) The manuscript is difficult to follow. I recommend preparing a diagram with the different neurons or signaling pathways (i.e. insulin) and how they affect each other.

We improved the data presentation by

a) adding a model showing the kinetics of memory formation in controls and mutants (Figure 2C)

b) a model explaining how the internal state is integrated into the formation of memory (Figure 7D).